# Towards a General Time Series Anomaly Detector with Adaptive Bottlenecks and Dual Adversarial Decoders

**Qichao Shentu**[1][*]**, Beibu Li**[1][*]**, Kai Zhao**[2]**, Yang Shu**[1][✉]**, Zhongwen Rao**[3]**,**
**Lujia Pan**[3]**, Bin Yang**[1]**, Chenjuan Guo**[1]
[1]East China Normal University, [2]Aalborg University, [3]Huawei Noah's Ark Lab
{qcshentu,beibul}@stu.ecnu.edu.cn, {yshu,cjguo,byang}@dase.ecnu.edu.cn,
{kaiz}@cs.aau.dk, {raozhongwen,panlujia}@huawei.com

## Abstract

Time series anomaly detection plays a vital role in a wide range of applications. Existing methods require training one specific model for each dataset, which exhibits limited generalization capability across different target datasets, hindering anomaly detection performance in various scenarios with scarce training data. Aiming at this problem, we propose constructing a general time series anomaly detection model, which is pre-trained on extensive multi-domain datasets and can subsequently apply to a multitude of downstream scenarios. The significant divergence of time series data across different domains presents two primary challenges in building such a general model: (1) meeting the diverse requirements of appropriate information bottlenecks tailored to different datasets in one unified model, and (2) enabling distinguishment between multiple normal and abnormal patterns, both are crucial for effective anomaly detection in various target scenarios. To tackle these two challenges, we propose a general time series anomaly **D**etector with **A**daptive Bottlenecks and **D**ual **A**dversarial Decoders (**DADA**), which enables flexible selection of bottlenecks based on different data and explicitly enhances clear differentiation between normal and abnormal series. We conduct extensive experiments on nine target datasets from different domains. After pre-training on multi-domain data, DADA, serving as a zero-shot anomaly detector for these datasets, still achieves competitive or even superior results compared to those models tailored to each specific dataset. The code is made available at https://github.com/decisionintelligence/DADA.

## 1 Introduction

With the continuous advancement of technology and the widespread application of various sensors, time series data is ubiquitous in many real-world scenarios (Anandakrishnan et al., 2017; Cook et al., 2020; Kieu et al., 2022; Tian et al., 2024; Pan et al., 2023). Effectively detecting anomalies in time series data helps to identify potential issues in time and takes necessary measures to ensure the normal operation of systems, thereby avoiding possible economic losses and security threats (Yang et al., 2021). For example, in the cyber-security field, timely detection of abnormal network traffic can prevent service interruptions (Dong et al., 2023a; Wang & Zhu, 2022); in the healthcare sector, detecting anomalies is crucial for preventing diseases (Wang et al., 2023b; Salem et al., 2021).

Anomaly detection methods based on deep learning (Xu et al., 2022; Su et al., 2019; donghao & wang xue, 2024) have recently achieved significant success due to their powerful representation capabilities for data. However, existing methods for detecting anomalies in time series data typically require constructing and training specific models for different datasets (Cook et al., 2020; Wang et al., 2023a; Campos et al., 2022; Yang et al., 2023). Although these methods show good performance on each specific dataset, their generalization ability across different target scenarios is limited (Zhang et al., 2023). In some scenarios, there are not

---

[*]Equal contribution.

enough data or resources for specific model training, such as those where data collection is impeded by data scarcity, time and labor costs, or user privacy concerns. Therefore, the feasibility of existing methods in practice may be greatly restricted. Aiming at this problem, we propose constructing a general time series anomaly detection (**GTSAD**) model. As shown in Figure 1, by pre-training the model on large time series data from multiple sources and domains, it is encouraged to learn anomaly detection capabilities from richer temporal information, which gives the potential to mitigate the overfitting of domain-specific patterns and learn patterns and models that are more generalizable. Then, the model can be efficiently applied to a wide range of downstream scenarios. However, constructing a general anomaly detection model still faces the following two major challenges.

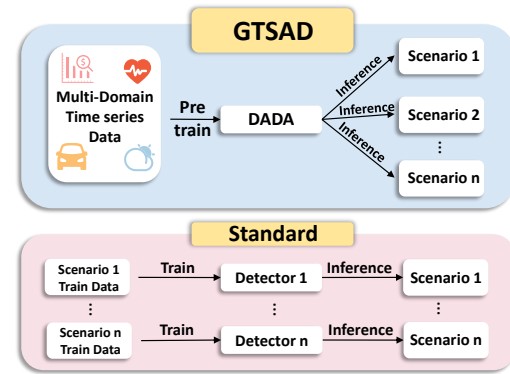

Figure 1: General time series anomaly detection (GTSAD) and standard methods training specific detectors for each scenario.

First, the large-scale time series data are from multiple sources and domains with different information densities, which raises the challenge of *meeting the diverse requirements of appropriate information bottlenecks tailored to different datasets in one unified model*. Existing anomaly detection methods emphasize accurately capturing the characteristics of normal data with autoencoder, due to the inherently scarce nature of anomalies (Chalapathy & Chawla, 2019). Information bottleneck is considered as a trade-off between compactly compressing the intrinsic information of the original data and high-fidelity reconstruction (Kawaguchi et al., 2023). A large bottleneck can reconstruct the original data accurately but may cause the model to fit unnecessary noise unrelated to normal patterns. On the contrary, a small bottleneck can compactly compress the intrinsic information but may result in the loss of diverse normal patterns and poor reconstruction (Wang et al., 2023a). Existing anomaly detection methods typically tune a fixed internal information bottleneck for each scenario, which is limited in inflexible representation capability and insufficient generalization ability when facing multi-domain time series data with significantly different data distributions, periodicity, and noise levels (Liu et al., 2024a), making it difficult to adapt well to downstream scenarios.

Second, the diverse manifestations of anomalies across multiple domains pose another significant challenge of *robust distinguishment between normal patterns and diverse anomaly patterns*. A model capable of general anomaly detection not only needs to have a clear understanding of specific normal patterns in time series data from different domains but also requires a clear delineation of decision boundaries between diverse normal and abnormal patterns. Existing anomaly detection methods are often based on the one-class classification assumption and only learn normal patterns in time series data for each specific domain, lacking explicit differentiation of the anomalies. Meanwhile, simply learning to discriminate normal and abnormal patterns specific to one domain is not generalizable enough, as the distributions of normal and abnormal patterns change across domains. As a result, they may struggle to distinguish between normal and abnormal patterns for general time series, where multi-domain data exist with diverse anomaly manifestations and the decision boundaries between normal and abnormal patterns are more complex.

In this paper, we propose a novel general time series anomaly Detector with Adaptive bottlenecks and Dual Adversarial decoders (DADA). **For the first challenge**, we consider the model's generalization ability from the perspective of a dynamic bottleneck and introduce the *Adaptive Bottlenecks* module to enhance the learning of normal time series patterns from multi-domain data. We employ a bottleneck to compress features into the latent space, the size of which can manifest different information densities for multi-domain data (Wang et al., 2023a). To meet the requirements of different information bottlenecks for divergent data, we establish a bottleneck pool containing various bottlenecks with different latent space sizes. Then, a data-adaptive mechanism is further proposed to enable the flexible selection of proper internal sizes based on the unique reconstruction requirements of the input data. **To address the second challenge**, we propose the *Dual Adversarial Decoders* module, which works with the encoder to enlarge the robust distinguishment between normal and anomaly patterns, where the normal decoder learns normal patterns for accurate reconstruction of

normal sequences, and the anomaly decoder learns different anomaly patterns from anomaly time series. By injecting noise that represents more common anomaly patterns, we prevent the model from overfitting domain-specific anomaly patterns that may vary across domains. We design an adversarial training mechanism for the encoder and the anomaly decoder on the reconstruction of abnormal series, which learns explicit decision boundaries between normal time series and common anomalies during the multi-domain training, thereby improving anomaly detection capability across different scenarios. Our main contributions are summarized as follows:

- We propose DADA, a novel General Time Series Anomaly **D**etector with **A**daptive Bottlenecks and **D**ual **A**dversarial Decoders. By pre-training on multi-domain time series data, we achieve the goal of "one-model-for-many", meaning a single model can perform anomaly detection on various target scenarios efficiently without domain-specific training.

- The proposed Adaptive Bottlenecks is the first to consider model's generalization ability from the perspective of the dynamic bottleneck in time series anomaly detection, which adaptively addresses the flexible reconstruction requirements of multi-domain data.

- The proposed Dual Adversarial Decoders explicitly amplify the decision boundaries between normal time series and common anomalies in an adversarial training way, which improves the general anomaly detection capability across different scenarios.

- Our model acts as a zero-shot time series anomaly detector that achieves competitive or superior performance on various downstream datasets, compared with state-of-the-art models trained specifically for each dataset.

## 2 RELATED WORK

**Time series anomaly detection.** Time series anomaly detection methods can mainly be categorized into none-learning, classical learning, and deep learning ones Zhao et al. (2022). Non-learning methods include density-based methods (Breunig et al., 2000) that determine which data points are abnormal by analyzing the density of the distribution of data points in the cluster, and similarity-based methods (Yeh et al., 2016) that series significantly dissimilar from most series are marked as abnormal. Classical learning methods (Liu et al., 2008) use a training dataset consisting of only normal data to classify whether the test data is similar to the normal data or not (Schölkopf et al., 1999). Deep learning methods mainly contain reconstruction-based and prediction-based methods. The reconstruction-based methods compress the raw input data and then attempt to reconstruct this representation, the abnormal data is often difficult to reconstruct correctly (Zhao et al., 2022; Wu et al., 2025). The classic approaches include the use of AEs in (Campos et al., 2022; Krizhevsky et al., 2012), VAEs in (Park et al., 2018; Li et al., 2021b) or GANs in (Li et al., 2019; Schlegl et al., 2019), and more recently, the highly successful utilization of transformer architectures in (Chen et al., 2022; Hu et al., 2024). The prediction-based methods (Pang et al., 2022) use past observations to predict the current value and use the difference between the predicted result and the real value as an anomaly criterion. Unlike these methods that require building a model for each dataset, our method mainly focuses on constructing a general time series anomaly detection model by pre-training on multi-domain time series data, which can be efficiently applied to various target datasets.

**Time series pre-training models.** Time series pre-training has received extensive attention. Some works develop general training strategies for time series. PatchTST (Nie et al., 2023) and SimMTM (Dong et al., 2023b) employ the BERT-style masked pre-training. InfoTS (Luo et al., 2023) and TS2Vec (Yue et al., 2022) utilize contrastive learning between augmented views of data. Additionally, Some works repurpose large language models to time series tasks (Gruver et al., 2023), such as GPT4TS (Zhou et al., 2023). Recently, novel time series pre-training models with zero-shot forecasting capabilities have emerged (Garza & Canseco, 2023; Woo et al., 2024; Liu et al., 2024a; Dooley et al., 2023; Liu et al., 2024b; Gruver et al., 2023), which demonstrate remarkable performance even without training on the target datasets. Nonetheless, there remains a dearth of time series anomaly detection models that support zero-shot. Existing pre-training methods for anomaly detection necessitate training on the target dataset. Different from previous methods, DADA pre-trained on multi-domain datasets is uniquely designed for time series anomaly detection, excelling in zero-shot anomaly detection across various target scenarios, thus filling this gap in the field.

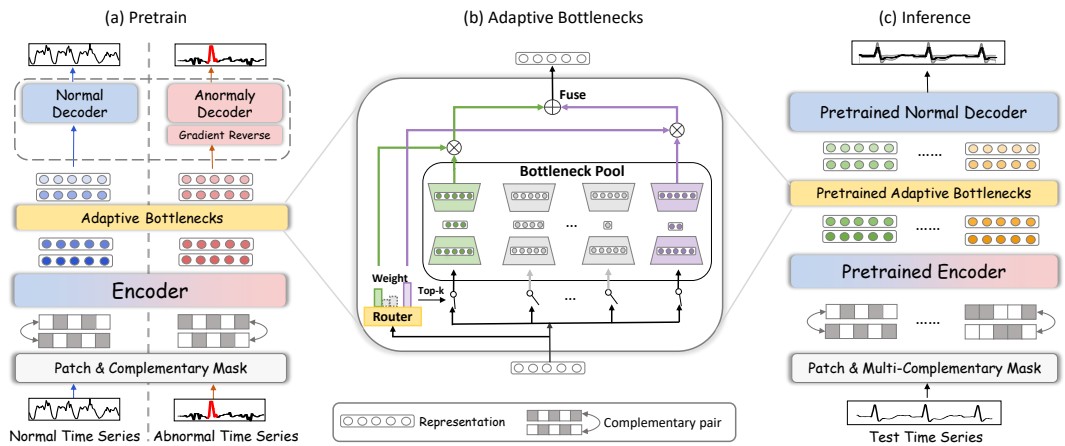

Figure 2: (a) The workflow during the pre-training stage. DADA mainly consists of Patch and Complementary Mask, Encoder, Adaptive Bottlenecks, and Dual Adversarial Decoders. (b) The structure of Adaptive Bottlenecks. (c) The workflow during the inference stage.

# 3 METHODOLOGY

Given a time series $\mathcal{D}_{\text{test}} = (\mathbf{x}_1, \mathbf{x}_2, \cdots, \mathbf{x}_T) \in \mathbb{R}^{T \times C}$ containing $T$ successive observations, where $C$ is the data dimensions. Time series anomaly detection outputs $\hat{\mathcal{Y}}_{\text{text}} = (y_1, y_2, \cdots, y_T)$, where $y_t \in \{0, 1\}$ denoting whether the observation $\mathbf{x}_t \in \mathbb{R}^C$ at a certain time $t$ is anomalous or not. In this paper, we focus on building a general time series anomaly detection model that is pre-trained on $M$ multi-domain time series datasets $\mathcal{D} = \{\mathcal{D}^{(i)}\}_{i=1}^M$, where each dataset $\mathcal{D}^{(i)} = \left( \mathbf{x}_1^{(i)}, \mathbf{x}_2^{(i)}, ..., \mathbf{x}_{T^{(i)}}^{(i)} \right) \in \mathbb{R}^{T^{(i)} \times C^{(i)}}$ has $C^{(i)}$-variates and $T^{(i)}$ time points. Then, the pre-trained model can be applied for anomaly detection on a target downstream dataset $\mathcal{D}_{\text{test}} \notin \mathcal{D}$ without any fine-tuning. Time series from different domains have varying numbers of variates. Thus, we use channel independence to extend our model across different domains (Nie et al., 2023; Yang et al., 2023; Goswami et al., 2024; Liu et al., 2024b; Wang et al., 2024). Specifically, a $C$-variate time series is processed into $C$ separate univariate time series.

## 3.1 OVERALL ARCHITECTURE

We propose DADA, a novel general time series anomaly **D**etector with **A**daptive bottlenecks and **D**ual **A**dversarial decoders. As shown in Figure 2, DADA employs a mask-based reconstruction architecture to model time series. During the pre-training stage, we utilize normal series, where no time points are labeled as anomalies, and abnormal series, which contains noise perturbation. Both normal and abnormal time series are concurrently input to DADA. The patch and complementary mask module maps each series to masked patch embeddings, which are fed into the encoder to extract temporal features. DADA employs adaptive bottlenecks for multi-domain time series data by dynamically selecting suitable bottleneck sizes. Finally, the dual adversarial decoders are employed to reconstruct the normal and abnormal series. During the inference stage, the anomaly decoder is removed, and DADA employs multiple complementary masks to produce multiple pairs of masked patch embeddings. The variance of the reconstructed values serves as the anomaly score, where the normal data can be stably reconstructed.

## 3.2 COMPLEMENTARY MASK MODELING

Time series mask modeling reconstructs masked parts using unmasked parts, to capture normal temporal dependencies from unmasked parts to masked parts. High reconstruction errors indicate anomalies that deviate from normal behavior. In this paper, we employ a complementary mask strategy by generating a pair of masked series with mutually complementary mask positions, which reconstruct each other to further capture comprehensive bi-directional temporal dependencies.

The complementary mask modeling consists of three steps. First, We use patch embedding to divide the univariate input time series into $P$ patches, each of dimension $d$, denoted as $\mathbf{X} \in \mathbb{R}^{P \times d}$. The division of time series data with patches has proven to be very helpful for capturing local information within each patch and learning global information among different patches, enabling the capture of complex temporal patterns (Nie et al., 2023; Yang et al., 2023). Then, we generate a masked series by randomly masking a portion of patches along the temporal dimension, formalized by $\mathbf{X}_{\mathrm{m}} = \mathbf{M} \odot \mathbf{X}$, where $\mathbf{M} \in \{0, 1\}^{P \times 1}$ is the mask and $\odot$ is the element-wise multiplication. At the same time, we generate another masked time series that complements $\mathbf{X}_{\mathrm{m}}$ according to the mask $\mathbf{M}$, formalized by $\bar{\mathbf{X}}_{\mathrm{m}} = (1 - \mathbf{M}) \odot \mathbf{X}$. Then, we reconstruct the masked parts, $\bar{\mathbf{X}}_{\mathrm{m}}$, based on $\mathbf{X}_{\mathrm{m}}$, and vice versa. This provides the model fully utilizing all data points to capture comprehensive bi-directional temporal dependencies. Finally, we combine the complementary reconstructed results. Denoting the process of reconstruction as $\mathrm{Recon}(\cdot) = \mathrm{Decoder}(\mathrm{AdaBN}(\mathrm{Encoder}(\cdot)))$, it passes its input through the Encoder, AdaBN, and Decoder to obtain the reconstruction results. The AdaBN and Decoder is detailed in Section 3.3 and Section 3.4. The final reconstructed results $\hat{\mathbf{X}}$ is calculated by:

$$\hat{\mathbf{X}} = (1 - \mathbf{M}) \odot \mathrm{Recon}(\mathbf{X}_{\mathrm{m}}) + \mathbf{M} \odot \mathrm{Recon}(\bar{\mathbf{X}}_{\mathrm{m}}). \tag{1}$$

## 3.3 ADAPTIVE BOTTLENECKS

As aforementioned, DADA is pre-trained on multi-domain time series datasets and applied to a wide range of downstream scenarios. This requires DADA to have the ability to learn generalizable representations from multi-domain time series datasets that show significant differences in data distribution, noise levels, etc. (Woo et al., 2024), and exhibit distinct preferences for the bottleneck. The size of the latent space can be considered as the internal information bottleneck of the model (Wang et al., 2023a). A large bottleneck may cause the model to fit unnecessary noise, while a small bottleneck may result in the loss of diverse normal patterns. The fixed bottleneck approaches used by existing methods (Campos et al., 2022; Su et al., 2019; Sakurada & Yairi, 2014) are poor in generalization ability and fail to detect anomalies across domains. Thus, we innovatively consider the model's generalization ability from the perspective of a dynamic bottleneck and propose an Adaptive Bottlenecks module (AdaBN) integrated with an adaptive router and a bottleneck pool, which dynamically allocates appropriate bottlenecks for multi-domain time series, enhancing the generalization ability of the model and enabling it to directly apply in various target scenarios.

**Bottleneck pool.** We compress the features into different latent spaces to achieve different information bottlenecks with different information densities. To accommodate the varying requirements of the multi-domain time series data, we have configured a pool of $B$ different sizes of bottlenecks $\mathrm{BN}_i(\cdot)$, $i = 1, 2, ..., B$, and each of them can compress the features into a different-sized latent space with dimension $d_i$. The masked time series input encoder can generate the corresponding representation $\mathbf{z} \in \mathbb{R}^{d_{\mathrm{r}}}$, where $d_{\mathrm{r}}$ denotes the dimension of representation. The process of each bottleneck can be formalized as:

$$\mathrm{BN}_i(\mathbf{z}) = \mathrm{UpNet}_i(\mathrm{DownNet}_i(\mathbf{z})). \tag{2}$$

$\mathrm{DownNet}_i(\cdot)$ represents the network which compresses the representation $\mathbf{z}$ into a latent space, $\mathbb{R}^{d_{\mathrm{r}}} \to \mathbb{R}^{d_{\mathrm{i}}}$, where $d_{\mathrm{i}}$ is the latent space dimension for the i-th bottleneck $\mathrm{BN}_i(\cdot)$ and $d_{\mathrm{i}} < d_{\mathrm{r}}$. $\mathrm{UpNet}_i(\cdot)$ restores the representation $\mathbf{z}$ from the latent space, $\mathbb{R}^{d_{\mathrm{i}}} \to \mathbb{R}^{d_{\mathrm{r}}}$.

**Adaptive router.** Employing all bottlenecks within the bottleneck pool indiscriminately will diminish model performance due to the mismatching between the information capacity of the latent space and the information densities of the data. An ideal model would dynamically allocate appropriate bottlenecks based on the intrinsic properties of the time series data. Therefore, we introduce the Adaptive Router, a dynamic allocation strategy that can flexibly select bottleneck size for each time series to address the flexible reconstruction requirements. As shown in Figure 2(b), the adaptive router employs a routing function to generate the selecting weights of each bottleneck from the bottleneck pool based on the representation. To avoid repeatedly selecting certain bottlenecks, causing the corresponding bottlenecks to be repeatedly updated while neglecting other potentially suitable bottlenecks, we add noise terms to increase randomness. We obtain the overall formula for the routing function $\mathrm{R}(\mathbf{z})$ as:

$$\mathrm{R}(\mathbf{z}) = \mathbf{z}\mathbf{W}_{\mathrm{router}} + \epsilon \cdot \mathrm{Softplus}(\mathbf{z}\mathbf{W}_{\mathrm{noise}}), \epsilon \sim \mathcal{N}(0, 1), \tag{3}$$

where $\mathbf{W}_{\text{router}}$ and $\mathbf{W}_{\text{noise}} \in \mathbb{R}^{d_{\text{r}} \times B}$ are learnable matrices for weights generation, $B$ is the number of bottlenecks and $\mathrm{Softplus}$ is the activation function. $\mathrm{R}(\mathbf{z})$ maps representation $\mathbf{z}$ to selection weights for $B$ bottlenecks. To encourage the model to update key bottlenecks, we select $k$ bottlenecks with the highest weights, denoting the set of their indexes as $\mathcal{K}$. Last, we assign higher attention weights to the output of more important bottlenecks and fuse their final output as:

$$\text{AdaBN}(\mathbf{z}) = \sum_{i \in \mathcal{K}} \frac{\exp\left(\mathrm{R}\left(\mathbf{z}\right)_i\right)}{\sum_{j \in \mathcal{K}} \exp\left(\mathrm{R}\left(\mathbf{z}\right)_j\right)} \mathrm{BN}_i\left(\mathbf{z}\right). \tag{4}$$

## 3.4 DUAL ADVERSARIAL DECODERS

**Reconstruction of normal time series.** As shown in Eq. (1), each input series $\mathbf{X}$ is reconstructed into result $\hat{\mathbf{X}}$. To achieve the goal of anomaly detection, we learn normal patterns from the reconstruction of normal time series. As shown in Figure 3, we use a feature extractor $G(\cdot; \theta_g)$, and a normal decoder $D_{\text{n}}(\cdot; \theta_n)$ with parameter $\theta_g$ and $\theta_n$ to denote the parts of the model used for reconstructing normal series. The $G$ includes the patch and complementary mask module, the encoder, and the adaptive bottlenecks. The decoder $D_{\text{n}}$ uses the output from the adaptive bottlenecks to reconstruct the series. We learn normal time series patterns by minimizing the reconstruction error of normal series:

$$\mathcal{L}_{\text{n}}(\theta_g, \theta_n) = \sum_{i=1}^{N_{\text{n}}} \|\mathbf{X}_{\text{n}}^{(i)} - \hat{\mathbf{X}}_{\text{n}}^{(i)}\|_2^2 = \sum_{i=1}^{N_{\text{n}}} \|\mathbf{X}_{\text{n}}^{(i)} - D_n(G(\mathbf{X}_{\text{n}}^{(i)}; \theta_g); \theta_n)\|_2^2, \tag{5}$$

where $N_{\text{n}}$ is the number of normal training series. $\mathbf{X}_{\text{n}}^{(i)}$ is the $i$-th normal series and $\hat{\mathbf{X}}_{\text{n}}^{(i)}$ is the corresponding reconstruction result.

**Reconstruction of abnormal time series.** However, merely learning to capture normal patterns is insufficient to detect anomalies in new scenarios. As a general time series anomaly detection model, the model will confront multi-domain data with diverse anomaly manifestations, and some patterns may only be specific to a certain domain and fail to generalize across domains, making the decision boundaries more complex. Thus, we need to explicitly enhance the model's ability to discriminate between normal patterns and some common abnormal patterns. We incorporate abnormal series with noise perturbations representing common anomaly patterns into the pre-training stage and propose an adversarial training stage that minimizes reconstruction errors for normal series while maximizing them for abnormal ones. Simply amplifying the reconstruction error for anomalies can

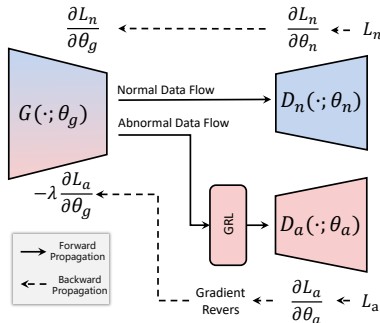

Figure 3: The forward and backward propagation of normal data and abnormal data.

confuse the feature extractor $G$ and fail to learn normal patterns. To address this, we introduce an anomaly decoder $D_{\text{a}}(\cdot; \theta_a)$ with parameter $\theta_a$ for adversarial training to constrain the model and encourage the feature extractor $G(\cdot; \theta_g)$ to learn normal pattern features. We seek the parameters $\theta_g$ of the feature extractor that maximizes the reconstruction loss of the abnormal data to make the representations contain abnormal information as little as possible, while simultaneously seeking the parameters $\theta_a$ of the anomaly decoder that minimizes the reconstruction loss of the abnormal data to learn anomaly patterns. The reconstruction loss for abnormal series can be formalized as:

$$\mathcal{L}_{\text{a}}(\theta_g, \theta_a) = \sum_{i=1}^{N_{\text{a}}} \|\left(\mathbf{X}_{\text{a}}^{(i)} - \hat{\mathbf{X}}_{\text{a}}^{(i)}\right) \odot \mathbf{Y}_{\text{a}}^{(i)}\|_2^2 = \sum_{i=1}^{N_{\text{a}}} \|\left(\mathbf{X}_{\text{a}}^{(i)} - D_{\text{a}}(G(\mathbf{X}_{\text{a}}^{(i)}; \theta_g); \theta_a)\right) \odot \mathbf{Y}_{\text{a}}^{(i)}\|_2^2, \tag{6}$$

where $N_{\text{a}}$ is the number of abnormal training series, $\mathbf{X}_{\text{a}}^{(i)}$ and $\mathbf{Y}_{\text{a}}^{(i)}$ denote the $i$-th abnormal series and its anomaly label. Our method does not strictly require labeled data. To reduce dependence on human-labeled data and to avoid overfitting to specific anomaly patterns, we generate anomalous data with more common anomaly patterns through anomaly injection. More details about anomaly injection are shown in Appendix A.3.

**Model training.**   As aforementioned, the overall optimization goal is to seek the parameters $\theta_g$ of feature extractor that maximize Eq. (6), while simultaneously seeking the parameters $\theta_a$ of anomaly decoder that minimize Eq. (6). In addition, we seek to minimize Eq. (5). As shown in Figure 3, we employ a Gradient Reversal Layer (GRL) (Ganin & Lempitsky, 2015) between $G$ and $D_a$. GRL alters the gradient from $D_a$, multiplies it by $-\lambda$, and passes it to $G$. That is to say, the partial derivatives of the loss for the encoder $\frac{\partial \mathcal{L}_a}{\partial \theta_g}$ is replaced with $-\lambda \frac{\partial \mathcal{L}_a}{\partial \theta_g}$, guiding parameters of different optimization objectives toward the desired direction for gradient descent, thereby avoiding the need for two separate optimizations. Finally, our objective can be formalized by:

$$(\theta_g^*, \theta_n^*) = \arg \min_{\theta_g, \theta_n} \mathcal{L}_n(\theta_g, \theta_n) - \mathcal{L}_a(\theta_g, \theta_a),$$
$$\theta_a^* = \arg \min_{\theta_a} \mathcal{L}_a(\theta_g, \theta_a). \tag{7}$$

**Anomaly criterion.**   During the inference stage, the model generates multiple pairs of complementary masked series for a single series and outputs multiple reconstructed series. Normal data points can be stably reconstructed, with the higher proximity of the reconstruction values. Conversely, reconstructions of abnormal data points are difficult and tend to be more unstable. Therefore, we utilize the variance of reconstructed values at the same time point as the anomaly score. Following existing works (Wang et al., 2023a; Su et al., 2019), we run the SPOT (Siffer et al., 2017) to get the threshold $\delta$, and a time point is marked as an anomaly if its anomaly score is larger than $\delta$.

# 4 EXPERIMENTS

## 4.1 EXPERIMENTAL SETTINGS

**Datasets.**   We employ a subset of the Monash (Godahewa et al., 2021) data hub and 12 datasets (Li et al., 2021a; Jacob et al., 2021; Ren et al., 2019; Roggen et al., 2010; Cui et al., 2016; Thill et al., 2020; Moody & Mark, 2001; Greenwald et al., 1990; Laptev et al., 2015) from anomaly detection task for pre-training. The pre-training datasets consist of approximately 400 million time points and intricately represent a wide range of domains. To demonstrate the effectiveness of our method, we evaluate five widely used benchmark datasets: SMD (Su et al., 2019), MSL (Hundman et al., 2018), SMAP (Hundman et al., 2018), SWaT (Mathur & Tippenhauer, 2016), PSM (Abdulaal et al., 2021), and NeurIPS-TS (including CICIDS, Creditcard, GECCO and SWAN) (Lai et al., 2021). We also evaluate UCR (Wu & Keogh, 2023) in the Appendix C.2. None of the downstream datasets for evaluation are included in the pre-training datasets. More details about the pre-training and evaluation datasets are shown in Appendix A.

**Baselines.**   We compare our model with 19 baselines for comprehensive evaluations, including the linear transformation-based models: OCSVM (Schölkopf et al., 1999), PCA (Shyu et al., 2003); the density estimation-based methods: HBOS (Goldstein & Dengel, 2012), LOF (Breunig et al., 2000); the outlier-based methods: IForest (Liu et al., 2008), LODA (Pevný, 2016); the neural network-based models: AutoEncoder (AE) (Sakurada & Yairi, 2014), DAGMM (Zong et al., 2018), LSTM (Hundman et al., 2018), CAE-Ensemble (Campos et al., 2022), BeatGAN (Zhou et al., 2019), OmniAnomaly (Omni) (Su et al., 2019), Anomaly Transformer (A.T.) (Xu et al., 2022), MEMTO (Song et al., 2024), DCdetector (Yang et al., 2023), D3R (Wang et al., 2023a), GPT4TS (Zhou et al., 2023), ModernTCN (donghao & wang xue, 2024), SensitiveHUE (Feng et al., 2024). To ensure a fair comparison, we adopt the same settings across all baselines, including the metrics and thresholding protocol, following common practices in anomaly detection.

**Metrics.**   Many existing methods use Point Adjustments (PA) to adjust the detection result. However, recent works have demonstrated that PA can lead to faulty performance evaluations (Huet et al., 2022). Even if only one point in an anomaly segment is correctly detected, PA will assume that the model has detected the entire segment correctly, which is unreasonable (Wang et al., 2023a). To overcome this problem, we use the affiliation-based F1-score (F1) (Huet et al., 2022), which has been widely used recently (Wang et al., 2023a; Yang et al., 2023). This score takes into account the average directed distance between predicted anomalies and ground truth anomaly events to calculate affiliated precision (P) and recall (R). Since the precision and recall are significantly influenced by thresholds, focusing solely on one of them fails to provide a comprehensive evaluation of the model,

Table 1: Results for five real-world datasets.

| Dataset | SMD | | | MSL | | | SMAP | | | SWaT | | | PSM | | |
|---|---|---|---|---|---|---|---|---|---|---|---|---|---|---|---|
| Metric | P | R | F1 | P | R | F1 | P | R | F1 | P | R | F1 | P | R | F1 |
| OCSVM | 66.98 | 82.03 | 73.75 | 50.26 | 99.86 | 66.87 | 41.05 | 69.37 | 51.58 | 56.80 | 98.72 | 72.11 | 57.51 | 58.11 | 57.81 |
| PCA | 64.92 | 86.06 | 74.01 | 52.69 | 98.33 | 68.61 | 50.62 | 98.48 | 66.87 | 62.32 | 82.96 | 71.18 | 77.44 | 63.68 | 69.89 |
| HBOS | 60.34 | 64.11 | 62.17 | 59.25 | 83.32 | 69.25 | 41.54 | 66.17 | 51.04 | 54.49 | 91.35 | 68.26 | 78.45 | 29.82 | 43.21 |
| LOF | 57.69 | 99.10 | 72.92 | 49.89 | 72.18 | 59.00 | 47.92 | 82.86 | 60.72 | 53.20 | 96.73 | 68.65 | 53.90 | 99.91 | 70.02 |
| IForest | 71.94 | 94.27 | 81.61 | 53.87 | 94.58 | 68.65 | 41.12 | 68.91 | 51.51 | 53.03 | 99.95 | 69.30 | 69.66 | 88.79 | 78.07 |
| LODA | 66.09 | 84.37 | 74.12 | 57.79 | 95.65 | 72.05 | 51.51 | 100.00 | 68.00 | 56.30 | 70.34 | 62.54 | 62.22 | 87.38 | 72.69 |
| AE | 69.22 | 98.48 | 81.30 | 55.75 | 96.66 | 70.72 | 39.42 | 70.31 | 50.52 | 54.92 | 98.20 | 70.45 | 60.67 | 98.24 | 75.01 |
| DAGMM | 63.57 | 70.83 | 67.00 | 54.07 | 92.11 | 68.14 | 50.75 | 96.38 | 66.49 | 59.42 | 92.36 | 72.32 | 68.22 | 70.50 | 69.34 |
| LSTM | 60.12 | 84.77 | 70.35 | 58.82 | 14.68 | 23.49 | 55.25 | 27.70 | 36.90 | 49.99 | 82.11 | 62.15 | 57.06 | 95.92 | 71.55 |
| BeatGAN | 74.11 | 81.64 | 77.69 | 55.74 | 98.94 | 71.30 | 54.04 | 98.30 | 69.74 | 61.89 | 83.46 | 71.08 | 58.81 | 99.08 | 73.81 |
| Omni | 79.09 | 75.77 | 77.40 | 51.23 | 99.40 | 67.61 | 52.74 | 98.51 | 68.70 | 62.76 | 82.82 | 71.41 | 69.20 | 80.79 | 74.55 |
| CAE-Ensemble | 73.05 | 83.61 | 77.97 | 54.99 | 93.93 | 69.37 | 62.32 | 64.72 | 63.50 | 62.10 | 82.90 | 71.01 | 73.17 | 73.66 | 73.42 |
| MEMTO | 49.69 | 98.05 | 65.96 | 52.73 | 97.34 | 68.40 | 50.12 | 99.10 | 66.57 | 56.47 | 98.02 | 71.66 | 52.69 | 83.94 | 64.74 |
| A.T. | 54.08 | 97.07 | 69.46 | 51.04 | 95.36 | 66.49 | 56.91 | 96.69 | 71.65 | 53.63 | 98.27 | 69.39 | 54.26 | 82.18 | 65.37 |
| DCdetector | 50.93 | 95.57 | 66.45 | 55.94 | 95.53 | 70.56 | 53.12 | 98.37 | 68.99 | 53.25 | 98.12 | 69.03 | 54.72 | 86.36 | 66.99 |
| SensitiveHUE | 60.34 | 90.13 | 72.29 | 55.92 | 98.95 | 71.46 | 53.63 | 98.37 | 69.42 | 58.91 | 91.71 | 71.74 | 56.15 | 98.75 | 71.59 |
| D3R | 64.87 | 97.93 | 78.04 | 66.85 | 90.83 | 77.02 | 61.76 | 92.55 | 74.09 | 60.14 | 97.57 | _74.41_ | 73.32 | 88.71 | 80.29 |
| ModernTCN | 74.07 | 94.79 | _83.16_ | 65.94 | 93.00 | 77.17 | 69.50 | 65.45 | 67.41 | 59.14 | 89.22 | 71.13 | 73.47 | 86.83 | 79.59 |
| GPT4TS | 73.33 | 95.97 | 83.14 | 64.86 | 95.43 | _77.23_ | 63.52 | 90.56 | _74.67_ | 56.84 | 91.46 | 70.11 | 73.61 | 91.13 | _81.44_ |
| DADA (zero shot) | 76.50 | 94.54 | **84.57** | 68.70 | 91.51 | **78.48** | 65.85 | 88.25 | **75.42** | 61.59 | 94.59 | **74.60** | 74.31 | 92.11 | **82.26** |

Table 2: Overall results for the NeurIPS-TS datasets.

| Dataset | CICIDS | | Creditcard | | GECCO | | SWAN | |
|---|---|---|---|---|---|---|---|---|
| Metric | F1 | AUC | F1 | AUC | F1 | AUC | F1 | AUC |
| A.T. | 34.71 | 49.00 | 65.14 | 52.55 | 64.27 | 51.60 | 33.67 | 44.74 |
| DCdetector | 40.02 | 53.95 | 58.28 | 42.36 | 66.18 | 45.38 | 14.42 | 43.48 |
| D3R | _67.79_ | 41.99 | 72.03 | 93.59 | **91.83** | 80.72 | 43.19 | 37.31 |
| ModernTCN | 51.74 | 65.33 | 73.80 | 95.55 | 90.18 | **95.95** | 46.45 | _52.63_ |
| GPT4TS | 54.00 | _67.91_ | _72.88_ | _95.58_ | 88.11 | 90.21 | _47.27_ | 51.93 |
| DADA (zero shot) | **73.49** | **69.33** | **75.12** | **95.73** | _90.20_ | _93.44_ | **71.93** | **53.29** |

and thus anomaly detection requires balancing these two metrics. Therefore, we pay more attention to the F1 score like widely used in anomaly detection Yang et al. (2023); Xu et al. (2022). We also employed the AUC_ROC (AUC) metric (Wang et al., 2023a; Campos et al., 2022). More implement details are shown in the Appendix B. All results are in %. The best ones are in bold and the second ones are underlined.

## 4.2 MAIN RESULTS

Each evaluation dataset has a training set and a test set. The standard baseline methods are trained on the training set and tested on the corresponding test set. Differently, our model DADA follows the zero-shot protocol. After the model is pre-trained on multi-domain datasets, it does not train on the downstream training set and is directly tested on the downstream test set. Note that all downstream evaluation datasets are not included in our pre-training datasets. As shown in Table 1, compared to the baselines directly trained with full data in each specific dataset, DADA, as a zero-shot detector, achieves state-of-the-art results in all five datasets, which demonstrates that DADA learned a general detection ability from a wide range of pre-training data, with clear distinguishment between multiple normal and anomalous patterns. We also evaluate our DADA on the NeurIPS-TS benchmark in Table 2, compared with the five recent methods that perform well as shown in Table 1. It can observed that these baselines fail to achieve consistent good results across different datasets. However, benefiting from pre-training on large-scale data, DADA achieves the best or most competitive results on all datasets, including the NeurIPS-TS datasets which are rich in anomaly types (Xu et al., 2022; Yang et al., 2023). It demonstrates the superior adaptability of DADA for different anomaly detection scenarios. More other metrics evaluation details can be found in Appendix C.

**Ablation study.** We further delve into the impact of individual components on the performance of DADA in zero-shot setting, as shown in Table 3. Firstly, we explore the impact of the AdaBN module. The first line results use a single fixed bottleneck for multi-domain datasets, eliminating the adaptive selection for various bottlenecks. When remove the AdaBN module, the outcomes exhib-

Table 3: Ablation studies. We study the influence of adaptive bottlenecks, adversarial training, and dual decoders. All results are in %, and the best ones are in bold.

| Dataset | | | SMD | | | MSL | | | SMAP | | | Avg |
| AdaBN | Adversarial | Dual Decoders | P | R | F1 | P | R | F1 | P | R | F1 | F1 |
| --- | --- | --- | --- | --- | --- | --- | --- | --- | --- | --- | --- | --- |
| ✗ | ✓ | ✓ | 71.13 | 93.61 | 80.84 | 62.79 | 91.74 | 74.55 | 58.91 | 84.10 | 69.29 | 74.45 |
| ✓ | ✗ | ✓ | 69.46 | 55.32 | 61.59 | 49.89 | 88.57 | 63.83 | 52.62 | 82.59 | 64.28 | 63.23 |
| ✓ | ✓ | ✗ | 75.77 | 87.82 | 81.35 | 64.82 | 84.32 | 73.29 | 61.50 | 80.27 | 69.64 | 74.76 |
| ✓ | ✗ | ✗ | 77.82 | 89.95 | 83.44 | 67.54 | 88.56 | 76.63 | 65.65 | 83.09 | 73.34 | 77.81 |
| ✗ | ✗ | ✗ | 78.60 | 74.96 | 76.74 | 65.20 | 72.69 | 68.74 | 55.22 | 88.57 | 68.03 | 71.17 |
| ✓ | ✓ | ✓ | 76.50 | 94.54 | **84.57** | 68.7 | 91.51 | **78.48** | 65.85 | 88.25 | **75.42** | **79.49** |

ited a decline of 5.04%. This emphasizes the criticality of dynamic bottlenecks for multi-domains pre-training. We also examine the influence of the dual adversarial decoders and its adversarial mechanism. The second line results directly maximize the reconstruction error of abnormal data, removing the adversarial mechanism. The third line results use a single decoder to reconstruct normal time series and abnormal time series, and the fourth line results simultaneously remove the adversarial mechanism and use a single decoder. We observe degradation in performance of w/o Adversarial by 16.26% (79.49%→63.23%) compared with DADA, and 7.89% (71.17%→63.23%) compared with the fifth line results, suggesting the needs for confrontation with the feature extractor, and when directly maximizing abnormal time series, the abnormal decoder can achieve this goal by generating arbitrary reconstruction results and undermine the encoder's ability to model normal patterns. Besides, employing a single decoder to handle both normal and abnormal time series causes zero-shot performance to degrade, instructing the necessity of dual decoders.

## 4.3 MODEL ANALYSIS

We analyze the effectiveness of adaptive bottlenecks and multi-domain pre-training and visualize the anomaly scores. We conducted additional analytical experiments, and the results are presented in Appendix D and Appendix E.

**Finetune with downstream data.** As depicted in Figure 4, we further demonstrate the results of fine-tuning DADA on downstream datasets under different data scarcities. After pre-training, the model develops strong generalization capabilities and exhibits promising zero-shot anomaly detection performance. By applying the model to a downstream dataset for further learning of domain-specific patterns, the model's detection ability can be further enhanced. After fine-tuning, the model's performance increases from 78.48% to 79.90% on MSL, and from 84.57% to 85.50% on SMD.

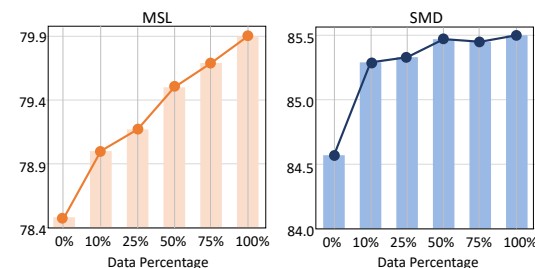

Figure 4: Results of fine-tuning DADA under different data percentage.

**Analysis on adaptive bottlenecks.** To validate the effectiveness of AdaBN, we remove the AdaBN module and instead employ a fixed bottleneck size. As shown in Figure 5(left), we observe that different bottleneck sizes exhibit different preferences for various downstream scenarios, while also introducing notable drawbacks in certain scenarios, e.g. the model with a bottleneck size of 32 achieves 74.81% on Creditcard, whereas only 67.56% on MSL. However, DADA, with adaptive bottlenecks dynamically allocating appropriate bottlenecks, consistently outperforms models utilizing a single bottleneck, which further substantiates the efficacy of our method. We also visualize the proportion of different adaptive bottleneck sizes that each dataset selects in Appendix E.5

**Analysis on multi-domain pre-training.** We further evaluate the baseline's zero-shot performance after multi-domain pre-training. Adopting the same setting as DADA, these models are first pre-trained extensively on multi-domain datasets, followed by zero-shot evaluation on the target datasets. From the presented Figure 5(right), we can observe that these baselines are incapable of effectively extracting generalization capability from multi-domain datasets, thus failing to deliver

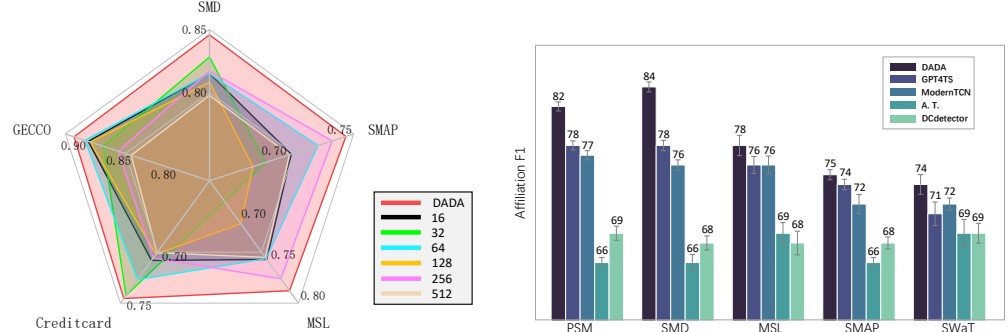

Figure 5: Model Analysis. (left) Comparison between DADA using AdaBN and using a single bottleneck. (right) Pre-training baseline model on multi-domain datasets and then evaluate them as a zero-shot detector on the target dataset.

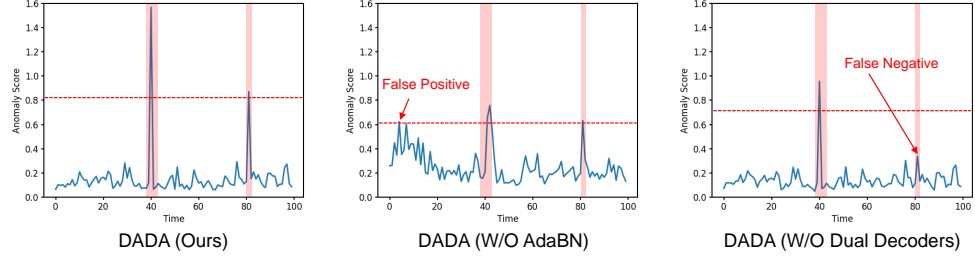

Figure 6: Visualization of the anomaly scores on PSM dataset. The pink intervals denote the ground truth anomalies, while the blue lines represent the anomaly scores produced by corresponding models. A red horizontal dashed line signifies the threshold. Any timestamp with an anomaly score exceeding this threshold is identified as an anomaly.

satisfactory results on the target dataset. This indicates that although multi-domain pre-training is important, simply using multi-domain datasets without proper methods still cannot obtain desired generalization ability. In contrast, DADA, equipped with its unique adaptive bottlenecks and dual adversarial decoders module tailored for the task of general anomaly detection, demonstrates great multi-domain pre-training and general detection ability.

**Visual analysis.** To further explain the efficacy of each module in DADA, we randomly sample time segments of length 100 for visualization. Figure 6 illustrates the anomaly scores at every timestamp within a given period and ground truth labels. After DADA removes AdaBN (W/O AdaBN), the model cannot adapt well to the normal pattern of multi-domain data and gets a higher false positive. After removing the dual decoders (W/O Dual Decoders), the model's decision boundary between normal and abnormal data becomes blurred and the false negative becomes higher. DADA (ours) shows that DADA not only achieves superior detection accuracy but also maintains a lower false alarm rate, a testament to its refined performance in time series anomaly detection.

## 5 CONCLUSION

This paper proposes a novel general time series anomaly detection model DADA. By pre-training on extensive multi-domain datasets, DADA can subsequently apply to a multitude of downstream scenarios without fine-tuning. We propose adaptive bottlenecks to meet the diverse requirements of information bottlenecks tailored to different datasets in one unified model. We also propose dual adversarial decoders to explicitly enhance clear differentiation between normal and abnormal patterns. Extensive experiments prove that as a zero-shot detector, DADA still achieves competitive or even superior results compared to those models tailored to each specific dataset.

## ACKNOWLEDGMENTS

This work was supported by National Natural Science Foundation of China (62372179, 62406112).

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

# A  DATASETS

## A.1  DATASETS FOR EVALUATION

We evaluate DADA and baselines on various datasets come from different domains, which can be broadly divided into spacecraft, servers, water treatment, finance, networking, and space weather: (1) **SMD** (Server Machine Dataset) collects resource utilization of computer clusters from an Internet company (Su et al., 2019). (2) **MSL** (Mars Science Laboratory dataset), collected by NASA, encompasses telemetry data reflecting the operational status of both sensors and actuators aboard the Martian rover (Hundman et al., 2018). (3) **SMAP** (Soil Moisture Active Passive dataset) is also collected by NASA and reflects soil moisture data from spacecraft monitoring systems (Hundman et al., 2018). (4) **SWaT** (Secure Water Treatment) collects sensor data from a continuously operating infrastructure (Mathur & Tippenhauer, 2016). (5) **PSM** (Pooled Server Metrics dataset) is collected from eBay Server Machines (Abdulaal et al., 2021). (6) **NeurIPS-TS** (NeurIPS 2021 Time Series Benchmark) is a dataset proposed by (Lai et al., 2021) and we use four sub-datasets: CICIDS, Creditcard, GECCO, SWAN with a variety of anomaly scenarios. (7) **UCR** Dataset is collected by Wu & Keogh (2023), which contains 250 sub-datasets. Each sub-datasets contains one dimension and one anomaly segment. For MSL and SMAP datasets, we only retain the first continuous dimension (Wang et al., 2023a). We summarize the datasets in Table 4.

Table 4: Details of benchmark datasets for evaluation. AR (anomaly ratio) represents the abnormal proportion of the whole dataset.

| Dataset | Domain | Dimension | Training | Validation | Test(labeled) | AR(%) |
|---|---|---|---|---|---|---|
| MSL | Spacecraft | 1 | 46,653 | 11,664 | 73,729 | 10.5 |
| PSM | Server Machine | 25 | 105,984 | 26,497 | 87,841 | 27.8 |
| SMAP | Spacecraft | 1 | 108,146 | 27,037 | 427,617 | 12.8 |
| SMD | Server Machine | 38 | 566,724 | 141,681 | 708,420 | 4.2 |
| SWaT | Water treatment | 31 | 396,000 | 99,000 | 449,919 | 12.1 |
| Creditcard | Finance | 29 | 113,922 | 28,481 | 142,404 | 0.17 |
| GECCO | Water treatment | 9 | 55,408 | 13,852 | 69,261 | 1.25 |
| CICIDS | Web | 72 | 68,092 | 17,023 | 85,116 | 1.28 |
| SWAN | Space Weather | 38 | 48,000 | 12,000 | 60,000 | 23.8 |
| UCR | Natural | 1 | 1,790,680 | 447,670 | 6,143,541 | 0.6 |

## A.2  DATASETS FOR PRE-TRAINING

We collect two datasets for pre-training, namely Anomaly Detection Data and Monash[+], each of them contains normal time series and abnormal time series. None of the downstream datasets for evaluation are included in the pre-training datasets. Firstly, we collect the Anomaly Detection Data from time series anomaly detection task. Table 5 lists subsets in Anomaly Detection Data, including ASD (Li et al., 2021a), Exathlon (Jacob et al., 2021), ECG (Moody & Mark, 2001), MITDB (Moody & Mark, 2001), OPP (Roggen et al., 2010), SVDB (Greenwald et al., 1990), GAIA[1], IOPS (Ren et al., 2019), MGAB (Thill et al., 2020), NYC (Cui et al., 2016), SKAB[2], YAHOO (Laptev et al., 2015). In the field of time series anomaly detection, datasets usually contain default normal training data and labeled test data with anomalies. We take the training data of this part as normal data, and the data with anomaly labels in test data as abnormal data.

To further enlarge our pre-training data, we build the Monash[+] dataset. The normal time series in Monash[+] comes from the Monash Prediction Library (Godahewa et al., 2021) as shown in Table 6, and the abnormal time series in Monash[+] are generated from Monash by anomaly injection. The anomaly injection is shown in Figure 7. These time series with different variate numbers and series lengths from multi-domains reflect the diverse application of the real world and the overall pre-training datasets are summarized in Table 7.

---

[1]https://github.com/CloudWise-OpenSource/GAIA-DataSet
[2]https://www.kaggle.com/dsv/1693952

Table 5: Anomaly Detection Data selected from multi-domain time series anomaly detection task. AR (anomaly ratio) represents the abnormal proportion of the whole dataset.

| Dataset | Domain | Dimension | Length | AR(%) |
|---------|--------|-----------|--------|-------|
| ASD | Application Server | 19 | 384,493 | 1.55 |
| Exathlon | Application Server | multi | 131,610 | 8.71 |
| ECG | Health | 1 | 9,681,300 | 4.7 |
| MITDB | Health | 1 | 13,650,000 | 3.44 |
| OPP | Health | 1 | 14,567,752 | 4.11 |
| SVDB | Health | 1 | 17,049,600 | 4.68 |
| GAIA | AIOps | 1 | 1,868,506 | 1.21 |
| IOPS | Web | 1 | 5,840,488 | 2.15 |
| MGAB | Mackey-Glass | 1 | 1,000,000 | 0.2 |
| NYC | Transport | 3 | 17,520 | 0.57 |
| SKAB | Machinery | 8 | 46,707 | 3.65 |
| YAHOO | Multiple | 1 | 565,827 | 0.62 |

Table 6: Multi-domain time series datasets from the Monash Prediction Library.

| Dataset | Domain | Frequency | Length |
|---------|--------|-----------|--------|
| Aus. Electricity Demand | Energy | Half Hourly | 1,155,264 |
| Wind | Energy | 4 Seconds | 7,397,147 |
| Wind Farms | Energy | Minutely | 172,178,060 |
| Solar Power | Energy | 4 Seconds | 7,397,223 |
| Solar | Energy | 10 Minutes | 7,200,857 |
| London Smart Meters | Energy | Half Hourly | 166,527,216 |
| Saugeen River Flow | Nature | Daily | 23,741 |
| Sunspot | Nature | Daily | 73,924 |
| Weather | Nature | Daily | 43,032,000 |
| KDD Cup 2018 | Nature | Daily | 2,942,364 |
| US Births | Nature | Daily | 7,305 |
| FRED_MD | Economic | Monthly | 77,896 |
| Bitcoin | Economic | Daily | 75,364 |
| NN5 | Economic | Daily | 87,801 |

## A.3 ANOMALY INJECTION

We use abnormal time series in the pre-training to explicitly enable robust distinguishment between normal and abnormal patterns with our dual adversarial decoders. Due to the scarcity of labeled data in the field of time series anomaly detection, we enrich abnormal samples by anomaly injection on the Monash dataset. These injected anomalies can be considered more common anomaly patterns. As shown in Figure 7, we generate an expanded collection of novel time series exhibiting diverse anomaly types and patterns, thereby enhancing our model's ability to distinguish between normal and abnormal time series. The injected anomalies cover point-wise anomalies and pattern anomalies. *hmirror* and *vmirror* alter the sequence trend by reversing the timestamps of sub-sequences or inverting the value magnitudes, respectively. *outlier* and *scale* generate outlier sequences or points by expanding (or reducing) the values of sub-sequences or specific time points. *Noise* and *pattern* simulate pattern anomalies by adding Gaussian noise to sub-sequences or replacing them with another sub-sequence. *Compress* and *stretch* modify the period length of the sequence through interval sampling or interpolation. We use the default values recommended in Schmidl et al. (2024), and empirical results show that these defaults produce reliable and robust anomaly injection outcomes.

For anomaly injection, we randomly select a subsequence within the original time series and an anomaly type. If the subsequence has not previously been injected with an anomaly, we inject this anomaly into it. We repeat this process until the anomaly ratio in the entire sequence reaches the pre-defined number.

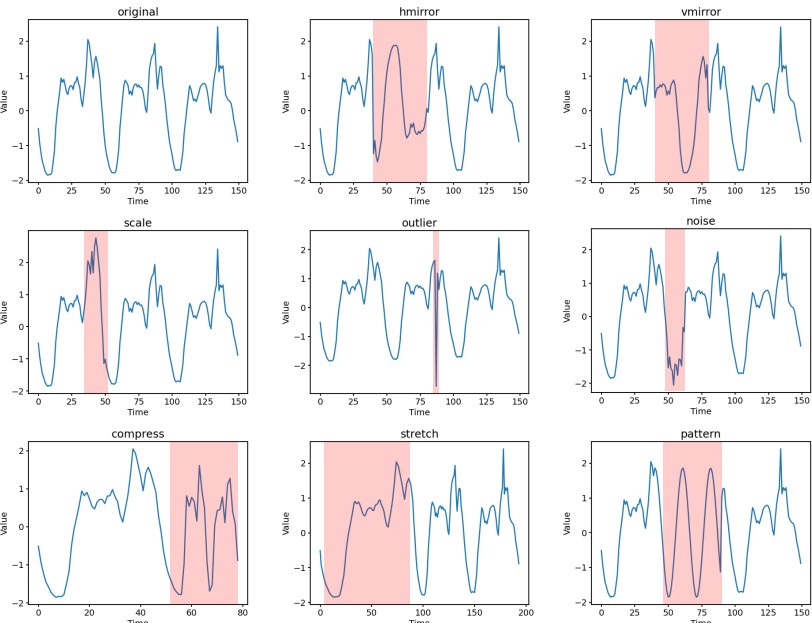

Figure 7: Anomaly injection. The blue lines represent the time series, while the pink areas intervals the anomalies we generated. We visualize the original time series and eight types of anomaly, including hmirror, vmirror, scale, outlier, noise, compress, stretch, and pattern.

Table 7: Description of pre-training datasets with normal data and abnormal data.

| Dataset | Normal Time Series | Abnormal Time Series |
|---|---|---|
| Anomaly Detection Data | Training set | Test set |
| Monash[+] | All of Moansh data | Generated from Monash by anomaly injection |

## B  IMPLEMENTATION DETAILS

We summarize all the default hyper-parameters as follows. We use a dilated CNN as an encoder with ten layers Yue et al. (2022). In the implementation phase, we run a sliding window with a window size of 100 and conduct anomaly detection using non-overlapping windows the same as existing works (Xu et al., 2022; Shen et al., 2020). Our patch size is 5. The bottleneck pool contains the sizes [16, 32, 64, 128, 192, 256] and the adaptive router selects $k = 3$ bottlenecks with the highest weights. The AdamW optimizer with an initial learning rate of $10^{-4}$ is used in pre-training. We set the batch size to 2048 with 5 epochs in pre-training. We do not use the "drop last" trick (Qiu et al., 2025b;a) during the testing phase to ensure a fair comparison, as advocated by TFB (Qiu et al., 2024). In the inference stage, we use 5 pairs of complementary masked series. After obtaining the anomaly score, we adopt commonly used SPOT (Siffer et al., 2017) following existing works (Wang et al., 2023a; Su et al., 2019) to get the threshold $\delta$ to determine whether each point is an outlier (Wang et al., 2023a). We conduct all experiments using Pytorch with NVIDIA Tesla-A800-80GB GPU.

## C  ADDITIONAL EXPERIMENTAL RESULTS

### C.1  MORE EVALUATION METRICS

In the realm of time series anomaly detection, there is a persistent debate regarding the most appropriate evaluation metrics to employ. This is due to the fact that different metrics can offer varying insights into the performance of a model, and the choice of metric can significantly influence the perceived effectiveness of an anomaly detection model. In our main text, we include the more reasonable Affiliated F1 and AUC_ROC scores. To further enhance the evaluation of our model, we also include the Volume Under the Surface (VUS) metric (Paparrizos et al., 2022), which takes

anomaly events into consideration based on the receiver operator characteristic curve. As shown in Table 8, DADA achieves good results on most metrics. GPT4TS, as a good baseline, demonstrates outstanding performance on the SMD dataset. However, its performance across different datasets is not consistent. For instance, there remains a noticeable gap compared to DADA on the PSM dataset. In contrast, DADA exhibits robust detection capabilities across all three datasets.

Table 8: Multi-metrics results.

| Dataset | Method | F1 | AUC_ROC | R_AUC_ROC | R_AUC_PR | VUS_ROC | VUS_PR |
|---|---|---|---|---|---|---|---|
| SMD | A.T. | 66.42 | 50.02 | 51.22 | 7.09 | 51.17 | 7.96 |
| | D3R | 78.02 | 53.34 | 62.89 | 11.20 | 61.69 | 11.01 |
| | GPT4TS | 83.13 | 71.15 | 77.27 | **17.68** | 76.79 | **17.45** |
| | ModernTCN | 83.16 | 70.21 | 77.54 | 16.28 | 77.07 | 15.96 |
| | DADA | **84.57** | **71.98** | **78.14** | 17.39 | **77.50** | 17.13 |
| MSL | A.T. | 66.49 | 34.05 | 38.65 | 10.25 | 38.90 | 10.41 |
| | D3R | 77.02 | 45.77 | 56.35 | 20.05 | 55.69 | 19.76 |
| | GPT4TS | 77.23 | 72.63 | 77.65 | 28.05 | 76.97 | 27.69 |
| | ModernTCN | 77.17 | 74.96 | 77.88 | 30.91 | 77.47 | 30.10 |
| | DADA | **78.48** | **75.15** | **77.99** | **31.05** | **77.48** | **30.50** |
| PSM | A.T. | 65.37 | 50.11 | 52.70 | 33.12 | 51.86 | 33.09 |
| | D3R | 80.29 | 50.22 | 52.93 | 39.62 | 52.18 | 39.31 |
| | GPT4TS | 81.44 | 58.94 | 65.16 | 46.54 | 64.66 | 45.99 |
| | ModernTCN | 79.59 | 58.46 | 65.50 | **47.48** | 64.80 | 46.68 |
| | DADA | **82.26** | **61.08** | **67.08** | 47.40 | **66.52** | **46.89** |

## C.2 EVALUATION ON THE UCR BENCHMARK

We evaluate DADA on UCR Anomaly Archive (Wu & Keogh, 2023), where the task is to find the position of an anomaly within the test set. To achieve this, we compute the anomaly scores for each time step in the test set and subsequently rank them in descending order. Following the approach of Timer (Liu et al., 2024b), if the time step with the $\alpha$ quantile hits the labeled anomaly interval in the test set, the anomaly detection is accomplished. Figure 8 presents the results of our evaluation. The left figure displays the number of datasets in which the model successfully detect anomalies at quantile levels of $3\%$ and $10\%$; a higher count indicates a stronger detection capability of the model. The right figure shows the quantile distribution and the average quantile across all UCR datasets. DADA has a lower average quantile compared to Timer, and which indicates better performance of anomaly detection.

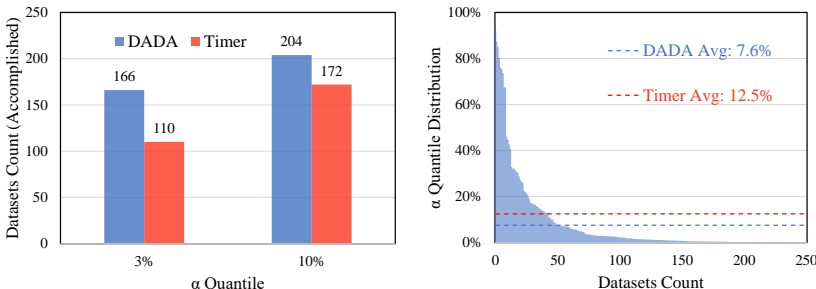

Figure 8: Downstream anomaly detection results for UCR benchmark

## D PARAMETER SENSITIVITY

We study the parameter sensitivity of the DADA. Figure 9 shows the model performance under different window sizes, $k$ bottlenecks with the highest weights that the adaptive router can select, and different patch sizes. Window size is a very important parameter for time series analysis. In a time series, a single point does not contain any information, so the window size determines the length of the sample context. Our experiments show that DADA is insensitive to window size and performs well across different window sizes. We then explore the influence of parameter $k$ in the adaptive bottlenecks. From the performance point of view, our model shows a relatively stable effect under different $k$, where we can achieve good performance even when selecting 1 bottleneck for each

sample. Finally, we study the performance under different patch sizes. Patch splitting has proven to be very helpful for capturing local information within each patch and learning global information among different patches. The results in Figure 9 show that DADA is robust under different patch sizes.

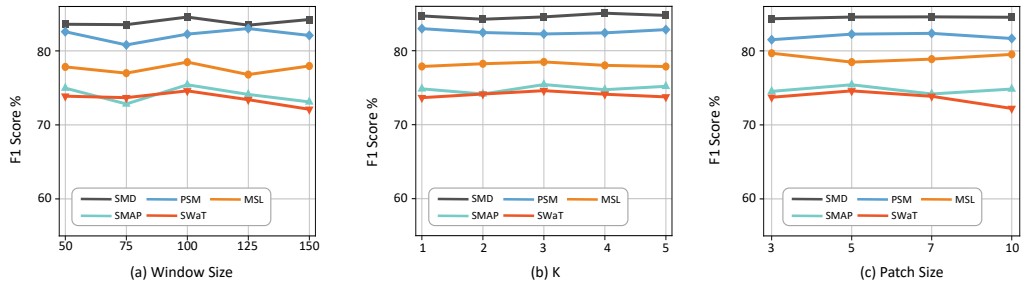

Figure 9: (a) Parameter sensitivity for sliding window size, (b) $k$ bottlenecks with the highest weights that the adaptive router can select, and (c) patch sizes (c).

# E ADDITIONAL MODEL ANALYSIS

## E.1 SHOW CASE

To provide a more intuitive illustration of how our model works, we employ anomaly injection to generate univariate time series featuring various anomalies, including outliers, horizontal mirroring, pattern shifts, and scale changes. As shown in Figure 10, our findings indicate that DADA can robustly and timely distinguish these anomalies from normal data points, which is crucial for real-world applications as it allows for the prompt prevention of potential losses. Moreover, leveraging the Dual Adversarial Decoders, if the model exhibits limitations in detecting some certain anomalies, its detection capabilities can be enhanced by incorporating the corresponding types of anomalies into the pre-training dataset.

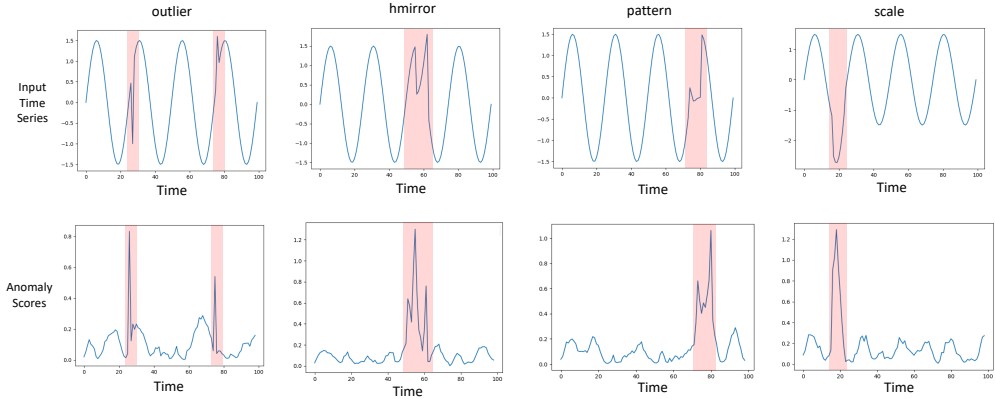

Figure 10: Visualization of learned criterion. The blue line in the first row represents the time series, and the blue line in the second row represents the anomaly scores obtained by the model, while the pink areas intervals the anomalies.

In Figure 11, we visualize the anomaly scores of GPT4TS, ModernTCN, and DADA on the PSM dataset. Additionally, we visualize the original sequences of four channels in the PSM dataset that contribute significantly to the anomalies. From the visualization results, it is clear that DADA effectively distinguishes the normal and abnormal samples, accurately identifying anomaly points. In contrast, GPT4TS and ModernTCN can not effectively increase the gap between the anomaly scores of normal points and anomalies, leading to many false positives.

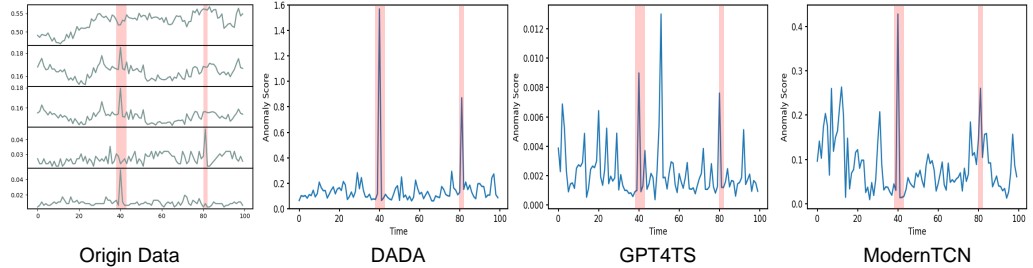

Figure 11: Visualization of anomaly score on PSM dataset. The green line in the origin data represents the original time series. The blue line represents the anomaly scores obtained by the model, while the pink areas intervals the anomalies.

## E.2 WITHOUT HUMAN-LABELED DATA

Our model does not strictly rely on human-labeled data. As mentioned in Appendix A, we can generate a variety of anomalies through anomaly injections. We conduct an experiment to validate this. We do not use any labeled data and only consider the anomaly injection method to pre-train a model (Without Label), and the zero-shot results under F1 are as shown in Table 9. It verifies that training our model without human-labeled supervised labels does not significantly impact the performance.

Table 9: Detection results without human-labeled data.

| Dataset | SMD | MSL | SMAP | SWAT | PSM |
|---|---|---|---|---|---|
| Metric | F1 | F1 | F1 | F1 | F1 |
| Without Label | 84.69 | 78.05 | 75.35 | 74.07 | 82.39 |
| DADA | 84.57 | 78.48 | 75.42 | 74.60 | 82.26 |

## E.3 COMPARISON WITH MOE

Mixture-of-Experts (MoE) (Jacobs et al., 1991; Shazeer et al., 2017) includes the gating network and multiple experts, each with the same structure. The gating network assigns different experts according to the characteristics of the data. In our AdaBN, each bottleneck in the bottleneck pool has a different hidden dimension. Each bottleneck compresses time series representation to a different hidden space to accommodate the varying requirements of the multi-domain time series data. As shown in table 10, to compare DADA with MoE, we replace all bottlenecks in the bottleneck pool with a consistent size of 256, which performs better compared to other hidden dimensions as shown in Figure 5. From Figure 5(left), we can see that MSL, SMAP, and SMD datasets are well-suited for a bottleneck size of 256, making it easier for MoE 256 to achieve good performance on these datasets. However, on datasets where the bottleneck size of 256 is less effective, such as SWaT, GECCO, and Creditcard, MoE 256's performance is not as good as AdaBN. It shows that MoE cannot solve the problem that different data require different bottlenecks in general time series anomaly detection. In contrast, DADA with AdaBN performs consistently across all datasets and shows significant improvements on SWaT, GECCO, and Creditcard compared to MoE 256, aligning with the motivation behind AdaBN's design that the model's bottleneck needs to be adaptively selected.

Table 10: Comparison of DADA and MoE. Single 256 replaces the AdaBN with a single bottleneck with size 256. The MoE 256 changes all bottlenecks in the bottleneck pool to the same size 256. The result is reported with Affiliated F1, and the best ones are in bold.

| Dataset | SMD | SMAP | MSL | SWaT | PSM | GECCO | Creditcard |
|---|---|---|---|---|---|---|---|
| Metric | F1 | F1 | F1 | F1 | F1 | F1 | F1 |
| single 256 | 81.69 | 74.37 | 77.01 | 69.93 | 80.15 | 85.12 | 70.96 |
| MoE 256 | 84.25 | 74.87 | 78.41 | 71.68 | 81.69 | 87.81 | 71.36 |
| DADA | **84.57** | **75.42** | **78.48** | **74.60** | **82.26** | **90.20** | **75.12** |

### E.4 INFERENCE TIME

In Table 11, we compare DADA with other models on the SMD dataset in terms of training time, inference time, and F1 score. Training time refers to the time required to train the model for one epoch on the target dataset, while inference time refers to the time required for a single forward pass on the testing set. We compare the baseline models A.T. and ModernTCN, as well as the pre-trained model GPT4TS. Additionally, we pre-train GPT4TS on multi-domain time series and include it in the comparison. It can be found that ModernTCN is advantageous in terms of inference time, but it still requires the training data to train the model on the target dataset. After pre-training on multi-domain datasets, GPT4TS (multi-domain) and DADA can perform zero-shot on target dataset. However, GPT4TS (multi-domain) requires a longer inference time and, due to not being specifically designed for anomaly detection, its detection performance is not ideal. In comparison, DADA has a certain advantage in inference time and can effectively perform zero-shot learning without target training data.

Table 11: Analysis on inference time.

| Methods | Training Time(s) | Inference Time(s) | F1 Score |
|---|---|---|---|
| A.T. | 782.5 | 0.093 | 66.42 |
| ModernTCN | 97.4 | **0.016** | 83.16 |
| GPT4TS | 934.9 | 0.113 | 83.13 |
| GPT4TS (multi-domain) | **0** | 0.113 | 78.00 |
| DADA | **0** | 0.081 | **84.57** |

### E.5 VISUALIZATION OF ADAPTIVE BOTTLENECKS

As shown in Figure 12(left), our visualization illustrates the proportion of different adaptive bottleneck sizes that each dataset selects and prefers. The figure highlights distinct preferences among datasets, indicating variability in their selection criteria. Notably, the SMAP and MSL datasets, sourced from NASA spacecraft monitoring systems, display a similarity in their data characteristics. This observation is further supported by our experimental results in Figure 12(left), showcasing consistent preferences for bottleneck sizes within these datasets.

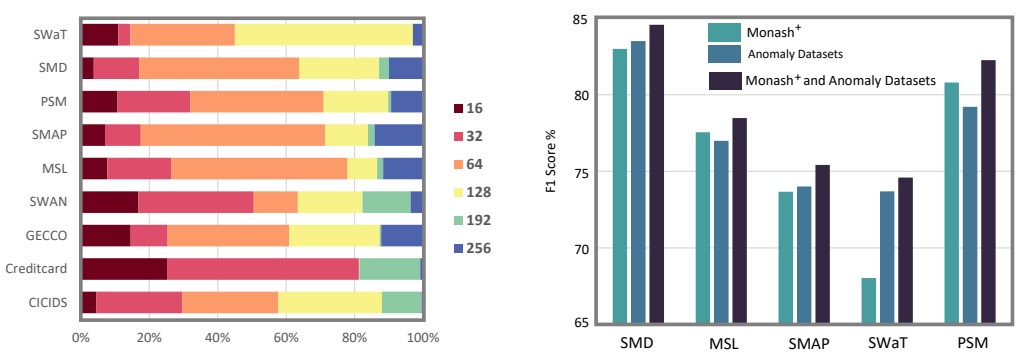

Figure 12: (left) Visualization of the proportion of different bottleneck sizes that each dataset prefers. (right) Pre-training with multi-domain datasets.

### E.6 PRE-TRAINING WITH MULTI-DOMAIN DATASETS

We investigated the impact of the pre-training datasets on our model's performance. As depicted in Figure 12(right), we employ the Monash[+] in Table 6 and Anomaly Detection Data in Table 5 individually for pre-training, followed by zero-shot evaluation on the SMD, MSL, SMAP, SWaT, and PSM datasets. Our model consistently exhibits great performance with different pre-training datasets.

Furthermore, using both datasets together to expand the domain scope of pre-training datasets enhances the breadth of information assimilated by our adaptive bottlenecks and dual adversarial decoders. Consequently, this enrichment contributes to further improvements in model performance.

### E.7 COMPLEMENTARY MASK ANALYSIS

The complementary mask is a way to enhance utilization during mask modeling. We mention this design for better reproducibility of our method. To further analyze it, we compare the ordinary masks that reconstruct the masked part from the unmasked part once with the complementary masks in Table 12. In terms of results, the complementary mask slightly outperforms the ordinary mask.

Table 12: Comparison between ordinary mask and complementary mask.

| Dataset | SMD | MSL | SMAP | SWaT | PSM |
|---|---|---|---|---|---|
| Metric | F1 | F1 | F1 | F1 | F1 |
| Ordinary mask | 84.25 | **78.51** | 75.03 | 74.32 | 81.87 |
| Complementary mask | **84.57** | 78.48 | **75.42** | **74.60** | **82.26** |

We further analyze the complementary mask rate's effect on anomaly detection performance. Since we use complementary masks, when data is masked 30%, its complementary part is masked 70%. Therefore, we analyze the effects of mask rates ranging from 10% to 50%. As shown in Table 13, we find that maintaining a complementary mask ratio closer to 50% yields better performance. When the mask ratio is too high, the model's reconstruction becomes very difficult, leading to poorer model performance.

Table 13: Analysis on complementary mask ratio.

| Dataset | SMD | SMAP | MSL | SWaT | PSM | Avg |
|---|---|---|---|---|---|---|
| Metric | F1 | F1 | F1 | F1 | F1 | F1 |
| 0.1 / 0.9 | 82.41 | 77.82 | 70.71 | 63.57 | 75.55 | 74.00 |
| 0.2 / 0.8 | 83.62 | 78.21 | 70.69 | 64.12 | 79.94 | 75.32 |
| 0.3 / 0.7 | 83.98 | 77.98 | 74.17 | 70.22 | 81.86 | 77.64 |
| 0.4 / 0.6 | 84.34 | 78.12 | **75.51** | 74.57 | **82.37** | 78.98 |
| 0.5 / 0.5 | **84.57** | **78.48** | 75.42 | **74.60** | 82.26 | **79.07** |

### E.8 ANOMALY CONTAMINATION IN TRAINING SAMPLE

Existing methods for time series anomaly detection typically assume that the training data are normal. Exploring the impact of potential anomalies in training data on the model is indeed an interesting research direction, but a major challenge is the lack of labels in training data to identify potential anomalies. To explore this impact, we additionally inject anomalies into our pre-training normal data at rates of 0.1%, 5%, and 10%, and treat these data as if they were normal. As shown in Table 14, the model's performance is affected to some extent as the anomaly contamination ratio increases. Since 10% is already much higher than the potential contamination in the actual training set, this result is acceptable and our method is generally robust to some anomaly contamination.

### E.9 VISUALIZATION OF PRE-TRAINING LOSS CURVES

As shown in Figure 13, we present the pre-training loss curves for the abnormal and normal parts, denoted as Loss_norm and Loss_abnorm. As training progresses, Loss_norm steadily declines, indicating improved reconstruction of normal data. In contrast, Loss_abnorm initially decreases but then rises slowly, stabilizing later, and is always clearly higher than Loss_norm. This is the expected effect of the adversarial training mechanism, as shown Eq. 7, where the Encoder maximizes the abnormal loss, and the anomaly decoder minimizes the abnormal loss. Eventually, Loss_abnorm

Table 14: Analysis on anomaly contamination in pre-training normal data.

| Dataset | SMD | MSL | SWaT | SMAP | Avg |
|---------|-----|-----|------|------|-----|
| Metric | F1 | F1 | F1 | F1 | F1 |
| 0% | **84.57** | **78.48** | **74.60** | 75.42 | **78.26** |
| 0.1% | 84.50 | 78.09 | 74.04 | **75.56** | 78.05 |
| 5% | 83.27 | 77.57 | 73.59 | 74.55 | 77.25 |
| 10% | 82.95 | 76.53 | 73.42 | 73.69 | 76.64 |

stabilizes at a steady value, indicating that the balance between the Encoder and the abnormal decoder has been reached and normal and abnormal data can be clearly distinguished. Therefore, the entire training process is relatively stable.

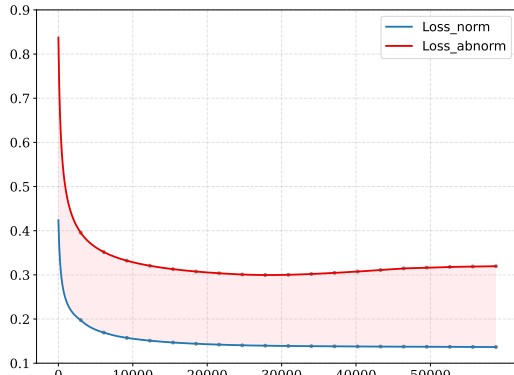

Figure 13: Pre-training loss curves for abnormal and normal parts, denoted as Loss_norm and Loss_abnorm.

