# OpenReview forum: "Towards a General Time Series Anomaly Detector with Adaptive Bottlenecks and Dual Adversarial Decoders"
_ICLR.cc/2025/Conference — ICLR 2025 Poster_

### Official Review · Reviewer_EBSp · 2024-10-16

**Soundness:** 3
**Presentation:** 4
**Contribution:** 3
**Rating:** 6
**Confidence:** 4

**Summary:**

The author propose a solution for general time series anomaly detection, which is call time series anomaly Detector with Adaptive bottlenecks and Dual Adversarial decoders (DADA). This model is  pre-trained on multi-domain time series data. Two components called Adaptive Bottlenecks and Dual Adversarial Decoders is leveraged to enhance the model performance. The model achieves competitive performance on various datasets even compared with SOTA models trained specifically for each dataset.

**Strengths:**

+ The task on zero-shot time series anomaly detection is quite challenging, the author build a unified end-to-end solution and demonstrate great results.

+ The paper is well-organized, and it introduces the background and motivation of the proposed methods clearly. The figures also explain their method pretty well.

+ The paper demonstrate its experiment results on enough datasets.  The ablation studies on Sec 4.3 also makes the paper convincible.

**Weaknesses:**

- The difference between the pretrain dataset and the evaluation dataset is not listed. How you select those dataset?

- It will be great to specify some more experiment details. Are there any data preprocessing on the input data?

- What's the computation cost(overhead) of importing Adaptive Bottlenecks and Dual Adversarial Decoders? Please specify.

**Questions:**

- Can you list the data difference between the pretrain dataset and the evaluation dataset? How you select the pretrain dataset?

- Are there any data preprocessing on the input data?

- What's the computation cost(overhead) of importing Adaptive Bottlenecks and Dual Adversarial Decoders? Please specify.

---

> ### Author Response · Authors · 2024-11-20
> **Response to reviewer EBSp**
>
> We sincerely thank reviewer EBSp for the valuable comments and suggestions. We have revised our paper accordingly and the changes are marked in blue.
>
> **W1 & Q1: Difference between pre-training and evaluation datasets**
> We list our evaluation datasets in $\underline{\text{Appendix A.1}}$ and our pre-training datasets in $\underline{\text{Appendix A.2}}$.
> **The differences between the pre-training and evaluation datasets are** (1) They are non-overlapping. None of the datasets used for pre-training are included in the evaluation. (2) They come from diverse domains. They include datasets where the domains are similar but the datasets are different, as well as datasets where the domains are significantly different. This indicates that their data patterns are different. (3) Each evaluation dataset comes from a specific domain. The whole pre-train dataset contains multiple domains.
> **How we select the datasets**. We choose 10 commonly used datasets in the anomaly detection field as evaluation datasets to assess model performance comprehensively. For pre-training, we gather datasets from various domains within anomaly detection. Additionally, we use the large-scale Monash dataset to further supplement the pre-training data. $\underline{\text{Table 7 in Appendix A}}$ provides a detailed description of the overall composition of the pre-training datasets we use.
>
> **W2 & Q2: More experiment details**
> Thanks for your suggestion. $\underline{\text{In Appendix A}}$, we introduce the collection and processing methodology for our datasets, covering pre-training data, evaluation data, and anomaly injection. The model processes these data using a channel-independent approach. $\underline{\text{In Appendix A.3 (in the revised paper)}}$, we have added a more detailed explanation of the types of anomalies injected and a comprehensive description of the entire anomaly injection process. Furthermore, $\underline{\text{in our code repository}}$, we have included the relevant code and scripts for anomaly injection.
>
> **W3 & Q3: Computation cost**
> The overall time complexity of the model is O(Encoder + AdaBN + Decoders). We adopt a sparse design in AdaBN where data goes through only top-k bottlenecks, and each bottleneck in AdaBN is designed to be lightweight. Each bottleneck and decoder is implemented using a simple MLP, so O(Encoder)>>O(AdaBN), O(Decoders). Therefore, the time complexity is approximately O(Encoder). We calculate the time cost of a single forward pass for each important module. As shown in the table, the main computational cost of the model comes from the Encoder. Introducing AdaBN and Abnorm Decoder has a minor impact on the model, which is acceptable, considering the improvements in zero-shot performance.
>
> | Encoder | AdaBN | Abnorm Decoder | Norm Decoder |
> | :---: | :---: | :---: | :---: |
> | 38.253 ms | 8.363 ms | 1.331 ms | 1.014 ms |

---

> ### Author Response · Authors · 2024-11-25
> **Looking forward to your feedback**
>
> Dear Reviewer EBSp:
>
> We would like to express our sincere gratitude for your time in reviewing our paper and your valuable comments.
>
> Since the rebuttal period is nearing its end, we are wondering whether our response has sufficiently addressed your questions. If so, we would greatly appreciate it if you could consider updating the score to reflect this. If you have any additional suggestions, we are more than willing to engage in further discussions and make necessary improvements to the paper.
>
> Thank you once again for dedicating your time to enhancing our paper!
>
> All the best,
>
> Authors

---

> > ### Comment · Reviewer_EBSp · 2024-12-01
> >
> > Thanks for the reply, the response has addressed my concerns.

---

> > > ### Author Response · Authors · 2024-12-01
> > > **Thank you for your response**
> > >
> > > Dear Reviewer EBSp:
> > >
> > > We are sincerely grateful for your support of our paper and pleased to know that our response addresses your concerns. We would greatly appreciate it if you could consider raising the score to reflect this.
> > >
> > > Thank you again for your valuable comments and dedicated time to review our paper.
> > >
> > > All the best,
> > >
> > > Authors

---

> ### Author Response · Authors · 2024-12-03
>
> We sincerely thank you for the insightful reviews and valuable comments, which are highly instructive for further improving our paper.
>
> In this work, we propose the **first general time series anomaly detection model**, and enhance its generalization ability and detection performance by introducing novel perspectives. After pre-training, our method, as a **zero-shot** anomaly detector, achieves state-of-the-art performance compared to models trained specifically for each dataset.
>
> Since the rebuttal period is nearing its end, **we kindly ask you to consider raising the score if we have addressed the concerns,** which would provide us with a greater opportunity to present our work at the conference.
>
> Thank you once again for dedicating the time to enhancing our paper!

---

### Official Review · Reviewer_e5Ze · 2024-10-24

**Soundness:** 3
**Presentation:** 3
**Contribution:** 3
**Rating:** 6
**Confidence:** 4

**Summary:**

The paper proposes DADA method for general time series anomaly detection, which is pre-trained on extensive multi-domain datasets and perform anomaly detection on various target scenarios without domain-specific training. DADA proposes Adaptive Bottlenecks to addresses the flexible reconstruction requirements of multi-domain data. DADA also employs Dual Adversarial Decoders amplify the decision boundaries between normal time series and common anomalies in an adversarial training way, which improves the general anomaly detection capability across different scenarios. The paper conducts extensive experiments on nine target datasets from different domains, and demonstrate that DADA achieves superior results compared with state-of-the-art models trained specifically for each dataset.

**Strengths:**

The paper addresses the issue that each dataset in the field of time series anomaly detection requires a specifically trained model, which holds practical significance. The paper proposes a novel method DADA for zero-shot inference on the target domain after pre-training on multiple datasets from different domains. DADA proposes the Adaptive Bottlenecks to addresses the flexible reconstruction requirements of multi-domain data, along with dual adversarial decoders to enhance the clear differentiation between normal and abnormal patterns, showcasing a degree of originality. The paper has a clear structure and expression. In the experimental section of the paper, DADA demonstrates superiority over multiple state-of-the-art methods across nine datasets and provides extensive analysis to prove the necessity of each module, showcasing high quality.

**Weaknesses:**

1. The paper lacks an analysis of the complementary mask rate's effect on anomaly detection performance. For example, the author could add an experiment to demonstrate the changes in anomaly detection F1-score under various complementary mask rates and explain why this mask performs the best.

2. In the experimental section, Figure 4 shows that after fine-tuning, the model's performance improved from 78.48% to 79.90% on MSL and from 84.57% to 85.50% on SMD, an increase of about 1%. The data shows that the improvement of the model after fine-tuning is small.

3. In the ablation study, Table 3 shows that the model performs better when removing the AdaBN, Adversarial, and Dual Decoders modules compared to removing only the Adversarial module. The paper explains that directly maximizing the anomalous time series will increases confusion. Further explanation is needed to clarify why maximizing the anomalous time series leads to increased confusion.

4. There are labeling mistakes in the experimental results. For example, in Table 8, the R_AUC_PR value of DADA on the PSM dataset is 47.40, which is less than ModernTCN's 47.48, but 47.40 is bolded. In Table 2, the F1 metric column for the CICIDS dataset is missing a underline marking.

**Questions:**

1. The paper does not compare the inference time of DADA with the baselines. Compared to the baselines, does DADA have a shorter inference time?  In particular, when compared to other pre-trained models, does DADA have an advantage?

2. In the DADA model, the original data first undergoes complementary masking operations, and then features are extracted by the encoder. In subsection 3.2, it states that the masked portions of the original data input to the encoder have a value of 0, but there may also be 0 values in the non-masked portions of the original data. How does the encoder differentiate between the masked and non-masked portions?

---

> ### Author Response · Authors · 2024-11-20
> **Response to reviewer e5Ze**
>
> We sincerely thank reviewer e5Ze for the valuable comments and suggestions. We have revised our paper accordingly and the changes are marked in blue.
>
> **W1: Analysis of the mask rate's effect**
> Thanks for your suggestion. Since we use complementary masks, when data is masked 30%, its complementary part is masked 70%. Therefore, we analyze the effects of mask rates ranging from 10% to 50%. From the experimental results, we find that maintaining a complementary mask ratio closer to 50% yields better performance. When the mask ratio is too high, the model's reconstruction becomes very difficult; when the mask ratio is too low, it becomes too simple, leading to poorer model performance. We have added the experiment $\underline{\text{in Appendix E.10 (in the revised paper)}}$.
>
> |dataset|SMD|MSL|SMAP|SWaT|PSM|Avg|
> |-|-|-|-|-|-|-|
> |0.1 / 0.9|82.41|77.82|70.71|63.57|75.55|74.00|
> |0.2 / 0.8|83.62|78.21|70.69|64.12|79.94|75.32|
> |0.3 / 0.7|83.98|77.98|74.17|70.22|81.86|77.64|
> |0.4 / 0.6|84.34|78.12|**75.51**|74.57|**82.37**|78.98|
> |0.5 / 0.5 (DADA)|**84.57**|**78.48**|75.42|**74.60**|82.26|**79.07**|
>
> **W2: About the fine-tuning result**
> Thank you for pointing this out. This work focuses on learning strong generalization and anomaly detection capabilities from multi-domain data to create a general anomaly detection model. The good zero-shot results indicate that the model has learned strong generalization capabilities from multi-domain pre-training, which reduces the gains on target datasets. Additionally, the fine-tuned results are 2.5% higher than SOTA using the same training data, which is a significant improvement. We will also explore how to better fine-tune a general anomaly detection model for target datasets in our future work.
>
> **W3: Explanation of why maximizing the anomalous time series leads to increased confusion**
> Thank you for your suggestions and we have added the explanations in the Ablation study $\underline{\text{in Section 4.2 (in the revised paper)}}$.
> For normal time series, our model utilizes the encoder and the normal decoder to minimize reconstruction errors, thereby learning normal patterns. For anomalous time series data, the model passes it through the encoder and then to the anomaly decoder. The anomaly decoder aims to minimize the reconstruction loss for abnormal series, allowing it to learn common abnormal patterns. However, the encoder aims to maximize the reconstruction loss, thereby enhancing the model's ability to discriminate between normal patterns and common abnormal patterns, which is a key aspect of the adversarial training.
> As illustrated in Table 3, removing the adversarial mechanism leads to a scenario where both the encoder and the anomaly decoder directly maximize the abnormal reconstruction loss simultaneously. This results in the anomaly decoder being unable to learn meaningful knowledge about abnormal patterns, as it can achieve this goal simply by generating arbitrary reconstruction results. Directly maximizing the abnormal reconstruction loss does not teach the model anything and even adds unnecessary noise to the pre-training process, weakening the encoder's ability to model normal patterns. So we call it "confusion".

---

> ### Author Response · Authors · 2024-11-20
> **Response to reviewer e5Ze**
>
> **W4: Labeling mistakes**
> Thank you very much for your detailed review. We have corrected labeling mistakes in the revision.
>
> **Q1: Inference time**
> We compare DADA with other models on the SMD dataset in terms of training time, inference time, and F1 score. Training time refers to the time required to train the model for one epoch on the target training dataset, while inference time refers to the time required for a single forward pass on the target testing set. We compare the baseline models A.T. and ModernTCN, as well as the pre-trained model GPT4TS. Additionally, we pre-train GPT4TS on multi-domain time series and include it in the comparison.
> It can be found that ModernTCN is advantageous in terms of inference time, but it still requires training data to train the model on the target dataset.
> After pre-training on multi-domain datasets, GPT4TS (multi-domain) and DADA can perform zero-shot on the target dataset. However, GPT4TS (multi-domain) requires a longer inference time and, due to not being specifically designed for anomaly detection, its detection performance is not ideal.
> In comparison, DADA has a certain advantage in inference time and can effectively perform zero-shot learning without target training data. Thanks for your suggestion, and we have added this experiment $\underline{\text{in Appendix E.11}}$.
>
> |Methods|TrainingTime(s)|InferenceTime(s)|F1Score|
> |-|-|-|-|
> |A.T.|782.5|0.093|66.42|
> |ModernTCN|97.4|**0.016**|83.16|
> |GPT4TS|934.9|0.113|83.13|
> |GPT4TS (multi-domain)|**0**|0.113|78.00|
> |DADA|**0**|0.081|**84.57**|
>
> **Q2: How to differentiate between masked and non-masked portions**
> $\underline{\text{In Section 3.2 of the paper}}$, we omit the patch embedding process. In fact, after dividing the original sequence into patches, the sequence first undergoes embedding before being masked. This is consistent with the provided code. Points in the original sequence that are 0 are converted into high-dimensional vectors along with other points in the same patch. The mask operation sets the entire patch embedding to 0, distinguishing it from the original 0 points. We apologize for any confusion this may have caused. We have clarified the patch embedding process in the revised paper and revised the dimension of $\mathbf{X}$ in the paper to $P\times d$, where $d$ is the dimension of patch embedding.

---

> ### Author Response · Authors · 2024-11-25
> **Looking forward to your feedback**
>
> Dear Reviewer e5Ze:
>
> We would like to express our sincere gratitude for your time in reviewing our paper and your valuable comments.
>
> Since the rebuttal period is nearing its end, we are wondering whether our response has sufficiently addressed your questions. If so, we would greatly appreciate it if you could consider updating the score to reflect this. If you have any additional suggestions, we are more than willing to engage in further discussions and make necessary improvements to the paper.
>
> Thank you once again for dedicating your time to enhancing our paper!
>
> All the best,
>
> Authors

---

> > ### Comment · Reviewer_e5Ze · 2024-11-27
> > **Thank you for your response**
> >
> > Thank you for your response. I have read through the response and my score remains.

---

> > > ### Author Response · Authors · 2024-11-27
> > > **Thank you for your response**
> > >
> > > Dear Reviewer e5Ze:
> > >
> > > We are sincerely grateful for your support of our paper.
> > >
> > > With the rebuttal period extended by one week, we have more time to further refine our work. If you have any additional suggestions, we are more than willing to engage in further discussions and make necessary improvements to the paper. We would be very grateful for any opportunity to further improve the score of our paper.
> > >
> > > Thanks again for your response and recognition of our work.
> > >
> > > All the best,
> > >
> > > Authors

---

> ### Author Response · Authors · 2024-12-03
>
> We sincerely thank you for the insightful reviews and valuable comments, which are highly instructive for further improving our paper.
>
> In this work, we propose the **first general time series anomaly detection model**, and enhance its generalization ability and detection performance by introducing novel perspectives. After pre-training, our method, as a **zero-shot** anomaly detector, achieves state-of-the-art performance compared to models trained specifically for each dataset.
>
> Since the rebuttal period is nearing its end, **we kindly ask you to consider raising the score if we have addressed the concerns,** which would provide us with a greater opportunity to present our work at the conference.
>
> Thank you once again for dedicating the time to enhancing our paper!

---

### Official Review · Reviewer_e12v · 2024-10-30

**Soundness:** 3
**Presentation:** 3
**Contribution:** 3
**Rating:** 6
**Confidence:** 4

**Summary:**

This paper develops a novel general time series anomaly detection by pre-training on multi-domain time series.  The proposed adaptive bottleneck enables the general model to be applied to various target scenarios without requiring domain-specific training. To improve the general anomaly detection capability, the dual adversarial decoder is further employed to amplify the decision boundary.

**Strengths:**

1. The general time series anomaly detection plays a key role in the real application.
2. DADA employs the adaptive bottlenecks for multi-domain time series data by dynamically selecting suitable bottleneck sizes.
3. The effectiveness of both AdaBN and Dual Decoders has been demonstrated in the experiments.

**Weaknesses:**

1. If anomaly injection is used to establish a supervised setting, this implies that all original samples(some anomalies are included) are treated as normal samples. I suggest the author add an experiment to examine performance under varying levels of anomaly contamination in the training samples.
2. During model training, two optimization losses are utilized, along with the optimization approach and any challenges encountered in the optimization process. The visualizations of loss curves for the two losses are essential for further demonstration.

**Questions:**

1. Does "anomaly injection" refer to incorporating multiple types of anomaly patterns into each dataset? More details about the implementation of anomaly injection should be included in the paper since it is crucial for conducting the supervised setting.
2. How do you adapt various baseline methods pre-trained on multi-domain datasets for zero-shot evaluation if they are typically utilized in the standard form?
3. How can DADA be applied during the inference step to further fine-tune the target datasets for downstream tasks in the experiments?

---

> ### Author Response · Authors · 2024-11-20
> **Response to reviewer e12v**
>
> We sincerely thank reviewer e12v for the valuable comments and suggestions. We have revised our paper accordingly and the changes are marked in blue.
>
> **W1: Anomaly contamination in training sample**
> Thank you for your suggestions. We have added this experiment $\underline{\text{in Appendix E.8 (in the revised paper)}}$. Existing methods for time series anomaly detection typically assume that the training data are normal. Exploring the impact of potential anomalies in training data on the model is indeed an interesting research direction, but a major challenge is the lack of labels in training data to identify potential anomalies. To explore this impact, we additionally inject anomalies into our pre-training normal data at rates of 0.1%, 5%, and 10%, and treat these data as if they were normal. Our results show that the model's performance is affected to some extent as the anomaly contamination ratio increases. Since 10% is already much higher than the potential contamination in the actual training set, this result is acceptable and our method is generally robust to some anomaly contamination.
>
> | Contamination Rate | SMD | MSL | SWaT | SMAP |
> | :---: | :---: | :---: | :---: | :---: |
> | 0% | **84.57** | **78.48** | **74.60** | 75.42 |
> | 0.1% | 84.50 | 78.09 | 74.04 | **75.56** |
> | 5% | 83.27 | 77.57 | 73.59 | 74.55 |
> | 10% | 82.95 | 76.53 | 73.42 | 73.69 |
>
> **W2: About loss curve**
> Thanks for your suggestions. We have added the loss curves $\underline{\text{in Figure 13 in Appendix E.9 (in the revised paper)}}$. We present the pre-training loss curves for the abnormal and normal parts, denoted as Loss_norm and Loss_abnorm. As training progresses, Loss_norm steadily declines, indicating improved reconstruction of normal data. In contrast, Loss_abnorm initially decreases but then rises slowly, stabilizing later, and is always clearly higher than Loss_norm. This is the expected effect of the adversarial training mechanism, as shown $\underline{\text{in Eq.(7) in the paper}}$, where the Encoder maximizes the abnormal loss, and the anomaly decoder minimizes the abnormal loss. Eventually, Loss_abnorm stabilizes at a steady value, indicating that the balance between the Encoder and the abnormal decoder has been reached and normal and abnormal data can be clearly distinguished. Therefore, the entire training process is relatively stable.
>
> **Q1: More details about anomaly injection**
> Thank you for your suggestions. For anomaly injection, we randomly select a subsequence within the original time series and an anomaly type. If the subsequence has not previously been injected with an anomaly, we inject this anomaly into it. We repeat this process until the anomaly ratio in the entire sequence reaches the pre-defined number.
> We have added a more detailed explanation of the types of anomalies injected and a comprehensive description of the entire anomaly injection process $\underline{\text{in Appendix A.3 (in the revised paper)}}$. Additionally, we have added the relevant code and scripts for anomaly injection $\underline{\text{in our code repository}}$.
>
> **Q2: How can baseline pre-train and zero-shot**
> To handle the issue of pre-training on multi-domain datasets, we adopt the channel-independent approach for processing multi-domain data for these baseline models. These models continue to be trained according to their original training methods with multi-domain data and then directly tested on downstream datasets. For these models, the only change is that the training set of the downstream dataset is replaced by the multi-domain datasets, and they are not further trained on the downstream dataset, which allows them to perform zero-shot anomaly detection.
>
> **Q3: How can DADA fine-tune the target datasets**
> For fine-tuning, we discard the abnormal decoder and only use the normal decoder because the boundaries between normal and anomalous patterns are clearer with the target domain training data. We freeze the encoder and AdaBN modules and use $\underline{\text{Eq. (5) in the main text}}$ to fine-tune the normal decoder using the training set of the target dataset.

---

> > ### Comment · Reviewer_e12v · 2024-11-26
> >
> > Dear authors
> >
> > I truly appreciate the detailed clarification, which has addressed most of my concerns. I will maintain my positive score.

---

> > > ### Author Response · Authors · 2024-11-26
> > > **Thanks for your response**
> > >
> > > Dear Reviewer e12v:
> > >
> > > We are sincerely grateful for your support of our paper.
> > >
> > > With the rebuttal period extended by one week, we have more time to further refine our work. If you have any additional suggestions, we are more than willing to engage in further discussions and make necessary improvements to the paper. We would be very grateful for any opportunity to further improve the score of our paper.
> > >
> > > Thanks again for your response and recognition of our work.
> > >
> > > All the best,
> > >
> > > Authors

---

> ### Author Response · Authors · 2024-11-25
> **Looking forward to your feedback**
>
> Dear Reviewer e12v:
>
> We would like to express our sincere gratitude for your time in reviewing our paper and your valuable comments.
>
> Since the rebuttal period is nearing its end, we are wondering whether our response has sufficiently addressed your questions. If so, we would greatly appreciate it if you could consider updating the score to reflect this. If you have any additional suggestions, we are more than willing to engage in further discussions and make necessary improvements to the paper.
>
> Thank you once again for dedicating your time to enhancing our paper!
>
> All the best,
>
> Authors

---

> ### Author Response · Authors · 2024-12-03
>
> We sincerely thank you for the insightful reviews and valuable comments, which are highly instructive for further improving our paper.
>
> In this work, we propose the **first general time series anomaly detection model**, and enhance its generalization ability and detection performance by introducing novel perspectives. After pre-training, our method, as a **zero-shot** anomaly detector, achieves state-of-the-art performance compared to models trained specifically for each dataset.
>
> Since the rebuttal period is nearing its end, **we kindly ask you to consider raising the score if we have addressed the concerns,** which would provide us with a greater opportunity to present our work at the conference.
>
> Thank you once again for dedicating the time to enhancing our paper!

---

### Official Review · Reviewer_hvUu · 2024-11-02

**Soundness:** 2
**Presentation:** 3
**Contribution:** 3
**Rating:** 6
**Confidence:** 5

**Summary:**

Authors propose a General time series anomaly Detector with Adaptive Bottlenecks and Dual Adversarial Decoders (DADA), which enables flexible selection of bottlenecks based on different data and explicitly enhances clear differentiation between normal and abnormal series. The zero-shot capability is indeed impressive.

**Strengths:**

The writing is clear and easy to understand.
The pictures and tables in the paper are beautiful and clear.
The proposed method, DADA, can achieve zero-shot.

**Weaknesses:**

1. AdaBN has almost no difference from the previous moe, yet the author exaggerates their design. I think there is a question of contribution. Moreover, the author does not explain the difference from Moe in the main text nor add relevant references [a]. I think this is somewhat deceptive.
2. The role of AdaBN is not as great as the author claims. See the analysis in question 1. There is excessive exaggeration of contribution.
[a] Shazeer, Noam, et al. "Outrageously large neural networks: The sparsely-gated mixture-of-experts layer." arXiv preprint arXiv:1701.06538 (2017).

**Questions:**

1. In the ablation experiment, the result of the fourth row is the second highest. In my opinion, AdaBN should be basically equivalent to moe, except for the different dimensions. However, at this time, the model effect is basically close to the final model, which means that the main source of performance is not the specially designed optimization method. This makes me doubt the component innovation of the paper. At the same time, Table 10 in the appendix also supports this conclusion. The result of moe 256 is completely better than the result of the fourth row. This seriously indicates that AdaBN is likely to have a much smaller actual effect than Moe 256. Further, the innovation of AdaBN should be greatly discounted.
2. The experimental analysis lacks the analysis of COMPLEMENTARY MASK MODELING. The author takes this item as the key point of the method but lacks further research.
3. The visualization analysis does not state what dataset is used, nor is there visualization of the original data. Furthermore, there is a lack of visualization comparison with other algorithms, lacking the credibility of the experiment.
4. I hope the author can open source the generation method and generation script of Monash+ and provide a directly usable data set. At present, the explanations in the paper are completely insufficient.

---

> ### Author Response · Authors · 2024-11-20
> **Response to reviewer hvUu**
>
> We sincerely thank reviewer hvUu for the valuable comments and suggestions. We have revised our paper accordingly and the changes are marked in blue.
>
> **W1: Comparison between AdaBN and MoE**
> The differences between AdaBN and the previous MoE models go beyond just dimensional variations. Our experiments, as detailed $\underline{\text{in Table 10 in Appendix E.3}}$, demonstrate that AdaBN outperforms the previous MoE models. Thank you for your suggestion, and we have added the reference you suggested.
> **Different intuitions.** AdaBN is specifically designed for multi-domain pre-training in time-series anomaly detection, focusing on tailoring appropriate information bottlenecks for various datasets within a unified model. It ensures that our model can flexibly adjust the bottleneck size to adapt to the different reconstruction requirements of different data. MoE, however, focuses on enhancing the model's representation capabilities by increasing the number of parameters while avoiding additional computational overhead. MoE cannot solve the problem that different data require different bottlenecks in general time series anomaly detection.
> **Different structures.** AdaBN establishes a bottleneck pool containing various bottlenecks with different latent space sizes between the encoder and decoder. It selects an appropriate bottleneck size for the input and then performs a reconstruction. However, MoE replaces the block in the original model by employing multiple experts with identical structures. These multiple experts capture different features and then integrate them.
>
> **W2 & Q1: The effect of AdaBN**
>
> 1. **Explanation of Ablation Study**. We would like to clarify the misunderstanding. The comparison between the fourth row and the sixth row (final model) in the ablation study illustrates the impact of the Dual Adversarial Decoders. The average improvement of 1.68% is significant. The comparison between the first row, i.e. w/o AdaBN, and the sixth row, demonstrates the improvements brought by adding the AdaBN module. The average improvement of 5.04% is also clearly substantial.
> 2. **Experimental comparison between AdaBN and MoE**. Directly comparing the result of MoE 256 in Table 10 with the fourth row in Table 3 is unfair. MoE 256 includes both MoE and Dual Adversarial Decoders, whereas the fourth row only has AdaBN. This comparison does not fairly demonstrate whether MoE outperforms AdaBN. The MoE 256 should be compared with the final model DADA $\underline{\text{in Table 10}}$, and AdaBN outperforms MoE on all datasets.
> 3. **MoE does not always perform well**. $\underline{\text{In Table 10 in the appendix}}$, the performance of MoE is comparable with AdaBN on MSL, SMAP, and SMD datasets, but it is clearly worse than AdaBN on SWaT and PSM. From $\underline{\text{Figure 5(left) in the main text}}$, we can see that MSL, SMAP, and SMD datasets are well-suited for a bottleneck size of 256, making it easier for MoE 256 to achieve good performance on these datasets. However, on datasets where the bottleneck size of 256 is less effective, such as SWaT, MoE 256's performance is not as good as AdaBN. This is further validated by adding the MoE 256 results on the Creditcard and GECCO datasets. It shows that MoE cannot solve the problem that different data require different bottlenecks in general time series anomaly detection. In contrast, AdaBN performs consistently across all datasets and shows significant improvements on SWaT, GECCO, and Creditcard compared to MoE 256, aligning with the motivation behind AdaBN's design that the model's bottleneck needs to be adaptively selected.
>
> |  | SMD | SMAP | MSL | SWaT | PSM | GECCO | Creditcard |
> | --- | --- | --- | --- | --- | --- | --- | --- |
> | MoE 256 | 84.25 | 74.87 | 78.41 | 71.68 | 81.69 | 87.81 | 71.36 |
> | DADA | **84.57** | **75.42** | **78.48** | **74.60** | **82.26** | **90.20** | **75.12** |
>
> 4. **Overall, We conducted extensive experiments to demonstrate the effectiveness of AdaBN**. In the ablation study in $\underline{\text{Table 3}}$, AdaBN brought an average improvement of 5.04%. $\underline{\text{Figure 5(left)}}$ illustrates the preferences of different datasets for different-sized bottlenecks. $\underline{\text{Figure 12(left) in Appendix E.5}}$ illustrates that AdaBN indeed can adaptively select an appropriate bottleneck based on the data, further substantiating the importance of AdaBN.

---

> ### Author Response · Authors · 2024-11-20
> **Response to reviewer hvUu**
>
> **Q2: About complementary mask**
> The complementary mask is a way to enhance utilization of training samples during mask modeling and we did not claim it as a main technical contribution of our paper. We mention this design for better reproducibility of our method. To further analyze it, we have added a comparison between the ordinary masks that reconstruct the masked part from the unmasked part once and the complementary masks $\underline{\text{in Table 11 in Appendix E.7 (in the revised paper).}}$ In terms of results, the complementary mask slightly outperforms the ordinary mask.
>
> |  dataset  |  SMD  |  MSL  |  SMAP  |  SWaT  |  PSM  |
> | --- | --- | --- | --- | --- | --- |
> | Ordinary mask | 84.25 | **78.51** | 75.03 | 74.32 | 81.87 |
> | Complementary mask | **84.57** | 78.48 | **75.42** | **74.60** | **82.26** |
>
> **Q3: Visualization analysis**
> Thank you for your suggestions. We use the PSM dataset, and we have added it in the caption of $\underline{\text{Figure 6 (in the revised paper)}}$. Following your suggestion, we have added a visualization of the original data and a visualization comparison with other baselines in $\underline{\text{Figure 11 in Appendix E.1 (in the revised paper)}}$. From the visualization results, it is clear that DADA distinguishes anomalies from normal points more effectively than ModernTCN and GPT4TS, avoiding many false alarms.
>
> **Q4: Generation of Monash+**
> Thank you very much for your suggestions. We have updated the generation script for Monash+ $\underline{\text{in our code repository}}$. After downloading the data from the Monash dataset link provided in the repository we can generate  Monash+ with the script.
> $\underline{\text{In Appendix A.3}}$, we describe and visualize the anomaly injection. To further explain the anomaly injection, we have added a more detailed explanation of different types of anomalies injected and a more comprehensive description of the entire anomaly injection process $\underline{\text{in Appendix A.3 (in the revised paper).}}$

---

> ### Author Response · Authors · 2024-11-25
> **Looking forward to your feedback**
>
> Dear Reviewer hvUu:
>
> We would like to express our sincere gratitude for your time in reviewing our paper and your valuable comments.
>
> Since the rebuttal period is nearing its end, we are wondering whether our response has sufficiently addressed your questions. If so, we would greatly appreciate it if you could consider updating the score to reflect this. If you have any additional suggestions, we are more than willing to engage in further discussions and make necessary improvements to the paper.
>
> Thank you once again for dedicating your time to enhancing our paper!
>
> All the best,
>
> Authors

---

> > ### Comment · Reviewer_hvUu · 2024-11-28
> >
> > Thank you for your response; it has addressed some of my concerns, and I have adjusted my score.

---

> > > ### Author Response · Authors · 2024-11-29
> > > **Thanks for your response**
> > >
> > > Dear Reviewer hvUu:
> > >
> > > We are sincerely grateful for your support of our paper.
> > >
> > > With the rebuttal period extended by one week, we have more time to further refine our work. If you have any additional suggestions, we are more than willing to engage in further discussions and make necessary improvements to the paper. We would be very grateful for any opportunity to further improve the score of our paper.
> > >
> > > Thanks again for your response and recognition of our work.
> > >
> > > All the best,
> > >
> > > Authors

---

> ### Author Response · Authors · 2024-12-03
>
> We sincerely thank you for the insightful reviews and valuable comments, which are highly instructive for further improving our paper.
>
> In this work, we propose the **first general time series anomaly detection model**, and enhance its generalization ability and detection performance by introducing novel perspectives. After pre-training, our method, as a **zero-shot** anomaly detector, achieves state-of-the-art performance compared to models trained specifically for each dataset.
>
> Since the rebuttal period is nearing its end, **we kindly ask you to consider raising the score if we have addressed the concerns,** which would provide us with a greater opportunity to present our work at the conference.
>
> Thank you once again for dedicating the time to enhancing our paper!

---

### Official Review · Reviewer_WTbT · 2024-11-04

**Soundness:** 3
**Presentation:** 3
**Contribution:** 3
**Rating:** 6
**Confidence:** 3

**Summary:**

This paper proposes a new general time series anomaly detector with adaptive bottleneck and dual adversarial decoder. By pre-training on multi-domain time series data, anomaly detection is performed efficiently in different target scenarios without domain-specific training.
The overall writing of this article is good and the logic is clear。
This paper first proposes an adaptive bottleneck module to compress features into the latent space, reflecting the different information densities of multi-domain data. It solves the challenge of different requirements for appropriate information bottlenecks for different datasets.
This paper uses dual adversarial decoders to expand the robustness distinction between normal and abnormal modes, improving the general anomaly detection ability in different scenarios.

**Strengths:**

The overall writing of this article is good and the logic is clear。
This paper first proposes an adaptive bottleneck module to compress features into the latent space, reflecting the different information densities of multi-domain data. It solves the challenge of different requirements for appropriate information bottlenecks for different datasets.
This paper uses dual adversarial decoders to expand the robustness distinction between normal and abnormal modes, improving the general anomaly detection ability in different scenarios.

**Weaknesses:**

The adaptive router formula of the adaptive bottleneck module proposed in this paper lacks certain innovation, and its formula is similar to the routing function proposed in “PATHFORMER: MULTI-SCALE TRANSFORMERS WITH ADAPTIVE PATHWAYS FOR TIME SERIES FORECASTING”.
Some formulas in this paper lack clear explanations, which may cause some confusion when reading. For example, the reconstruction process proposed in the article is denoted as Recon(·), but there is a lack of detailed description of this process.

**Questions:**

I hope the author can explain the difference between the routing function proposed in "PATHFORMER: MULTI-SCALE TRANSFORMERS WITH ADAPTIVE PATHWAYS FOR TIME SERIES FORECASTING". At the same time, give a detailed explanation of the reconstruction process Recon(·).

---

> ### Author Response · Authors · 2024-11-20
> **Response to reviewer WTbT**
>
> We sincerely thank reviewer WTbT for the valuable comments and suggestions. We have revised our paper accordingly and the changes are marked in blue.
>
> **W1 & Q1: Differences between Pathformer and AdaBN**
>
> 1. **The motivation of the router is different.** The router in DADA is specifically designed for multi-domain pre-training in time-series anomaly detection, focusing on the importance of the bottleneck size for generalization capabilities. It ensures that our model can flexibly adjust the bottleneck size to adapt to the different reconstruction requirements of different time series data, capturing the essential patterns while discarding irrelevant noise. The router in Pathformer primarily focuses on the forecasting task within a single downstream dataset. It emphasizes that the dataset can be modeled at different scales through the use of a routing function to select various patch sizes.
>
> 2. **The mechanism and implementation of the router are different.** DADA selects an appropriate bottleneck size for the representation of time series and then passes the representation through the corresponding bottleneck for suitable reconstruction. Pathformer first selects different patch sizes for time series and divides it into different patches, and then feeds these patches into different transformer blocks for modeling at various scales.
>
> **W2 & Q2: Explanation of the reconstruction process**
> Thank you for your suggestions. We have added more detailed explanations $\underline{\text{in Section 3.2 (in the revised paper)}}$. The Recon(·) function passes its input through the Encoder, AdaBN, and Dual Decoders modules to obtain the reconstruction results. Since some elements of the input are masked as 0, the model attempts to reconstruct these masked elements back to their original values. It can be formalized as:
> $$
> \mathrm{Recon}(\mathbf{X}) = \mathrm{Decoder}(\mathrm{AdaBN}(\mathrm{Encoder}(\mathbf{X}))),
> $$
> which is the same as the equation $D(G(X))$ $\underline{\text{in Eq.(5) and Eq.(6)}}$ in the main text, where $ D $ is the Decoder, $ G $ is the AdaBN and Encoder, and $ \mathbf{X} $ is the input. When the input is normal data, we use the normal decoder $ D_n $; otherwise, the abnormal decoder $D_a$ is used.

---

> ### Author Response · Authors · 2024-11-25
> **Looking forward to your feedback**
>
> Dear Reviewer WTbT:
>
> We would like to express our sincere gratitude for your time in reviewing our paper and your valuable comments.
>
> Since the rebuttal period is nearing its end, we are wondering whether our response has sufficiently addressed your questions. If so, we would greatly appreciate it if you could consider updating the score to reflect this. If you have any additional suggestions, we are more than willing to engage in further discussions and make necessary improvements to the paper.
>
> Thank you once again for dedicating your time to enhancing our paper!
>
> All the best,
>
> Authors

---

> > ### Comment · Reviewer_WTbT · 2024-12-02
> >
> > Thank you for your positive response, which explained in detail the difference between the routing function proposed in the article and the routing function proposed in "PATHFORMER: MULTI-SCALE TRANSFORMERS WITH ADAPTIVE PATHWAYS FOR TIME SERIES FORECASTING", and further explained the motivation and mechanism of the routing function proposed in the article. At the same time, you answered my questions about some functions in the article in detail and added detailed explanations. I will keep my positive rating unchanged.

---

> > > ### Author Response · Authors · 2024-12-02
> > > **Thank you for your response**
> > >
> > > Dear Reviewer WTbT:
> > >
> > > We are sincerely grateful for your support of our paper and pleased to know that our response addresses your concerns. Thank you again for your valuable comments and dedicated time to review our paper.
> > >
> > > All the best,
> > >
> > > Authors

---

> ### Author Response · Authors · 2024-12-03
>
> We sincerely thank you for the insightful reviews and valuable comments, which are highly instructive for further improving our paper.
>
> In this work, we propose the **first general time series anomaly detection model**, and enhance its generalization ability and detection performance by introducing novel perspectives. After pre-training, our method, as a **zero-shot** anomaly detector, achieves state-of-the-art performance compared to models trained specifically for each dataset.
>
> Since the rebuttal period is nearing its end, **we kindly ask you to consider raising the score if we have addressed the concerns,** which would provide us with a greater opportunity to present our work at the conference.
>
> Thank you once again for dedicating the time to enhancing our paper!

---

### Meta-Review · Area_Chair_Mr8p · 2024-12-21

**Metareview:**

The work aims to learn a generalist model for zero-shot time series anomaly detection. The model is built with Adaptive Bottlenecks and Dual Adversarial Decoders. It is trained on multi-domain datasets and then tested directly on different datasets.

**Strengths**
- Paper is well written [WTbT, hvUu, e5Ze, EBSp]
- The studied setting is practically sound [e12v, e5Ze]
- The model combining adaptive information bottleneck and dual adversarial decoders is novel [WTbT, e5Ze, EBSp]
- The model shows effective zero-shot TSAD performance across various downstream datasets [WTbT, hvUu, e12v, e5Ze, EBSp]


**Weaknesses**
- The adaptive bottleneck module itself lacks novelty [WTbT, hvUu]
- Lack of analysis on part of equations, modules or the empirical results [WTbT, hvUu, e5Ze]
- Lack of investigation into the performance under varying level of anomaly contamination [e12v]
- Some key implementation details of the proposed method, the evaluation protocol, the competing methods are missing [e12v, EBSp]
- The model has limited improvement if it is fine-tuned on the target dataset [e5Ze]
- Lack of computational time analysis [e5Ze, EBSp]

**Additional Comments On Reviewer Discussion:**

The work receives five reviews in total, and all reviewers interacted with the authors during author-reviewer discussion.

There are a number of strengths pointed out by multiple reviewers, including impressive clarity, practical setting, novel model, and zero-shot TSAD performance.

The authors also did a great rebuttal and managed to address most of the concerns from the reviewers, resulting in at least two increased ratings from weak rejects to weak accepts. As a result, the five reviewers are consistently positive towards the work.

As far as I understand, one thing not raised by the reviewers but could be important to consider is that there have been some zero-shot anomaly detection methods on data other than time series data, such as on image data and probably on other data types. I'm not sure whether this work is motivated/inspired by these methods, but it'd be added value if the related work section could discuss such research progress.

In summary, the work explores a new yet practical setting, presents a novel method that helps address the challenges in the new setting, and shows promising zero-shot results on multiple downstream datasets. There are some cons with the work, but the overall quality is above ICLR acceptance bar.

---

### Decision · Program_Chairs · 2025-01-22

Accept (Poster)